# Outlier-Robust Sparse Mean Estimation for Heavy-Tailed Distributions

**Ilias Diakonikolas**
University of Wisconsin-Madison
`ilias@cs.wisc.edu`

**Daniel M. Kane**
University of California, San Diego
`dakane@cs.ucsd.edu`

**Jasper C.H. Lee**
University of Wisconsin-Madison
`jasper.lee@wisc.edu`

**Ankit Pensia**
University of Wisconsin-Madison
`ankitp@cs.wisc.edu`

## Abstract

We study the fundamental task of outlier-robust mean estimation for heavy-tailed distributions in the presence of sparsity. Specifically, given a small number of corrupted samples from a high-dimensional heavy-tailed distribution whose mean $\mu$ is guaranteed to be sparse, the goal is to efficiently compute a hypothesis that accurately approximates $\mu$ with high probability. Prior work had obtained efficient algorithms for robust sparse mean estimation of light-tailed distributions. In this work, we give the first sample-efficient and polynomial-time robust sparse mean estimator for heavy-tailed distributions under mild moment assumptions. Our algorithm achieves the optimal asymptotic error using a number of samples scaling logarithmically with the ambient dimension. Importantly, the sample complexity of our method is optimal as a function of the failure probability $\tau$, having an *additive* $\log(1/\tau)$ dependence. Our algorithm leverages the stability-based approach from the algorithmic robust statistics literature, with crucial (and necessary) adaptations required in our setting. Our analysis may be of independent interest, involving the delicate design of a (non-spectral) decomposition for positive semi-definite matrices satisfying certain sparsity properties.

## 1 Introduction

One of the most fundamental problem setups in statistics is as follows: given $n$ i.i.d. samples drawn from an unknown distribution $P$ chosen arbitrarily from some known distribution family $\mathcal{P}$, infer some particular property of $P$ from the data. This generic model captures a range of statistical problems of interest, for example, parameter estimation (such as the mean and (co)variance of $P$), as well as hypothesis testing. While long lines of work have given us a deep understanding on the statistical and computational possibilities and limits on these problems, these results are not always applicable in real-world settings due to (i) modelling issues, that the underlying distribution $P$ might not actually be in the known family $\mathcal{P}$ but only being close to it, and (ii) the fact that the $n$ samples supplied might be corrupted, for example by nefarious actors in high-stakes applications [ABH+72].

The field of *robust statistics* aims to design estimators and testers that can tolerate up to a *constant* fraction of corrupted samples, independent of the potentially high dimensionality of the data [Tuk60, HR09]. Classical works in the field have identified and resolved the statistical limits of problems in this setup, both in terms of constructing estimators and proving impossibility results [Yat85, DL88, DG92, HR09]. However, the proposed estimators were not computationally efficient, often requiring exponential time to compute either in the number of samples or the number of dimensions [HR09].

36th Conference on Neural Information Processing Systems (NeurIPS 2022).

A recent line of work, originating in the computer science community, has developed the subfield of *algorithmic* robust statistics, aiming to design estimators that not only attain tight statistical guarantees, but are also computable in polynomial time. This line of research has provided computationally and statistically efficient estimators in a variety of problem settings (e.g., mean estimation, covariance estimation, and linear regression) under different assumptions (e.g., the distribution might be assumed to be (sub-)Gaussian, or can be heavy tailed); see [DK19] for a recent survey of results.

The focus of this paper is the robust mean estimation problem under sparsity constraints on the mean vector. Sparsity is an important structural constraint that is both relevant in practice, especially in the face of increasing dimensionality of modern data, and extensively studied for statistical estimation (see, e.g., the books [HTW15, EK12, van16]). In the specific context of robust sparse mean estimation, prior works have studied the case where the underlying distribution has light-tails, e.g., sub-exponential tails [BDLS17, DKK+19b, CDG+, DKK+22]. In particular, the case of a spherical Gaussian distribution is now rather well-understood both in terms of the optimal information-theoretic estimation error, as well as the conjectured *computational-statistical tradeoff* — namely, that there is a gap between the statistical performance of computationally efficient and inefficient estimators [DKS17, BB20]. In this work, we initiate the investigation of robust sparse mean estimation for *heavy-tailed* distributions, under only mild moment assumptions. Our main result is the first computationally efficient robust mean estimator in the heavy-tailed setting which leverages sparsity to reduce sample complexity from depending polynomially on the dimensionality to a logarithmic dependence. Importantly, our algorithm also achieves the optimal dependence on the failure probability $\tau$ as it tends to 0; see the next two subsections for further discussion.

## 1.1 Problem Setup

We first define the input contamination model before formally stating the statistical problem.

**Definition 1.1** (Strong Contamination Model). *Given a corruption parameter $\epsilon \in (0, 1/2)$ and a distribution $P$ on uncorrupted samples, an algorithm takes samples from $P$ with $\epsilon$-contamination as follows: (i) The algorithm specifies the number $n$ of samples it requires. (ii) $n$ i.i.d. samples from $P$ are drawn but not yet shown to the algorithm. (iii) An arbitrarily powerful adversary then inspects the entirety of the $n$ i.i.d. samples, before deciding to replace any subset of $\lceil \epsilon n \rceil$ samples with arbitrarily corrupted points, and returning the modified set of $n$ samples to the algorithm.*

Define the $\ell_{2,k}$-norm of a vector $v$, denoted by $\|v\|_{2,k}$, as the $\ell_2$-norm of the largest $k$ entries of a vector $v$ in magnitude. The goal is to estimate the mean vector in this sparse norm.

**Problem 1.** *Fix a corruption parameter $\epsilon \in (0, 1/2)$, error parameter $\delta > 0$, failure probability $\tau \in (0, 1)$, and distribution family $\mathcal{D}$ over $\mathbb{R}^d$. Suppose we have access to $\epsilon$-contaminated samples drawn from an unknown distribution $P \in \mathcal{D}$ with mean $\mu$. The problem is to compute an estimate $\hat{\mu}$ such that $\|\hat{\mu} - \mu\|_{2,k}$ is upper bounded by error $\delta$ with probability at least $1 - \tau$ over $n$ samples. The goal is then to give an estimator with the minimal sample complexity $n(k, \epsilon, \delta, \tau)$.*

The above problem is slightly more general than sparse mean estimation in the following sense. To estimate a $k$-sparse mean vector $\mu$ to error $\delta$, it suffices (see [CDG+, Lemma 3.2]) to: 1) get an estimate $\tilde{\mu}$ with $\|\tilde{\mu} - \mu\|_{2,k} \leq \delta/3$, and 2) round $\tilde{\mu}$ to the $k$ entries with the largest magnitude, and zero out all the other entries. The result of our paper solves robust mean estimation in the $\ell_{2,k}$ norm.

A key aspect of robust statistics is that, depending on the distribution family $\mathcal{D}$ we consider, the above problem is generally not solvable for all error parameters $\delta > 0$. This work focuses on sparse mean estimation for *heavy-tailed* distributions, where a commonly used model for heavy-tailedness is imposing only the mild assumption that the covariance is bounded by $I$, without any further tail assumptions (see Section 1.4 for more discussion). Even when $d = 1$ and even when there are infinitely many samples [DK19], it is known that in the heavy-tailed setting, the minimum $\delta$ achievable is in the order of $\sqrt{\epsilon}$. This immediately implies the same lower bound of $\Omega(\sqrt{\epsilon})$ for the minimum achievable $\delta$ in Problem 1.

Before discussing the algorithmic results in this paper, we first state known information-theoretic bounds on the sample complexity that applies to all estimators, efficient or not, for Problem 1 on distributions with covariance bounded by $I$, and for $\delta = \Theta(\sqrt{\epsilon})$.

**Fact 1.2** (Information-theoretic sample complexity: computationally-inefficient). *In Problem 1, for the distribution family $\mathcal{D}_2$ which is the set of distributions with covariance $\Sigma \preceq I$, and for $\delta = \Theta(\sqrt{\epsilon})$,*

*we have that $n(k, \epsilon, \delta, \tau) \asymp (k \log(d/k) + \log(1/\tau))/\epsilon$. That is, all algorithms need these many samples and there exists a (computationally-inefficient) estimator with this sample complexity. The upper bound is from [Dep20a, PBR20], and the lower bound follows from [LM19b] even in the absence of outliers (see also Footnote 2 in [DL22a]) and even when we restrict to the distribution family $\mathcal{D}_{Gaussian}$ which is the set of the Gaussian distributions with identity covariance.*

An interesting aspect of robust sparse mean estimation is that there is a conjectured statistical-computational tradeoff, namely that efficient algorithms require a qualitatively larger sample complexity than inefficient algorithms. There is evidence (in the form of SQ lower bounds and reduction-based hardness) that all efficient algorithms have a quadratically-worse dependence on $k$, that is, even for constant $\epsilon, \delta, \tau$, and $\mathcal{D}_{Gaussian}$ being identity-covariance Gaussians in Problem 1, the sample complexity of all efficient algorithms is at least $\tilde{\Omega}(k^2)$, as opposed to $\tilde{O}(k)$ in Fact 1.2. See [DKS17, BB20] for a detailed discussion.

Both the information-theoretic bound and the conjectured computational lower bound serve as benchmarks for our algorithm to match.

## 1.2 Our Result

We now state the results of this paper, before discussing in the next section our algorithmic approach and also the assumptions required in the results.

Our main result is a computationally efficiently robust mean estimator in the $\ell_{2,k}$ norm, with performance matching the conjectured computational-statistical tradeoff, under the standard heavy-tailed assumption that the covariance $\Sigma \preceq I$ and the additional mild assumption that the $4^{\text{th}}$ moment is bounded in all axis directions. In the following (and the rest of the paper), we use the notation $a \gg b$ for $a, b \in \mathbb{R}$ to mean there exists a sufficiently large constant $c$ with $a \geq cb$.

**Theorem 1.3** (Our result: computationally efficient). *Let $\epsilon \in (0, \epsilon_0)$ for small constant $\epsilon_0 > 0$. Let $P$ be a multivariate distribution over $\mathbb{R}^d$, where the mean and covariance of $P$ are $\mu$ and $\Sigma$ respectively. Suppose $\Sigma \preceq I$ and further suppose that for all $j \in [d]$, $\mathbb{E}[(X_j - \mu_j)^4] = O(1)$. Then, there is an algorithm such that, on input (i) the corruption parameter $\epsilon$, (ii) the failure probability $\tau$, (iii) the sparsity parameter $k$, and (iv) $T$, an $\epsilon$-corrupted set of $n \gg (k^2 \log d + \log(1/\tau))/\epsilon$ i.i.d. samples from $P$, it outputs $\widehat{\mu}$ satisfying $\|\widehat{\mu} - \mu\|_{2,k} = O(\sqrt{\epsilon})$ with probability $1 - \tau$ in $\text{poly}(n, d)$ time.*

Phrased in a slightly different language, when our estimator is given a sufficiently large number $n$ of $\epsilon$-corrupted samples, it outputs an estimate $\hat{\mu}$ satisfying $\|\widehat{\mu} - \mu\|_{2,k} = O\left(\sqrt{\frac{k^2 \log d}{n}} + \sqrt{\epsilon} + \sqrt{\frac{\log(1/\tau)}{n}}\right)$ with probability $1 - \tau$. We note also the guarantees of our algorithm remain the same under a weaker assumption on $\Sigma$: we need only $\|\Sigma\|_{\mathcal{X}_k} \leq 1$ ($\|\cdot\|_{\mathcal{X}_k}$ defined in Definition 1.4) instead of spectral norm. Informally, the $\mathcal{X}_k$ norm of a square matrix $A$ is a convex relaxation of finding the maximum of $v^\top A v$ over $k$-sparse vectors $v$. See Theorem D.12 for the stronger version of the main result, which assumes only that $\|\Sigma\|_{\mathcal{X}_k} \leq 1$.

As outlined above, the dependence of our sample complexity result on $k$ is tight with respect to the conjectured lower bound for efficient algorithms, and its dependence on $\tau$ and $\epsilon$ are also tight with respect to the information-theoretic lower bounds, even in the Gaussian case. In terms of the smallest achievable asymptotic error (even given infinitely many samples), we show in Lemma E.1 that, even after adding the mild axis-wise $4^{\text{th}}$ moment assumption in Theorem 1.3, the asymptotic error remains lower bounded by $\Omega(\sqrt{\epsilon})$ when $k$ is sufficiently large. The restriction on $k$ is very mild, and covers most parameter regimes of interest.

We also note that the sample complexity has a dependence on the failure probability that is $\log 1/\tau$, and importantly, this is an additive term in the complexity instead of multiplicative. This additive dependence was non-trivial to achieve even in the optimal rates for heavy-tailed mean estimation in the non-robust (and non-sparse) setting. See the [LM19a] survey for a more detailed discussion.

## 1.3 Our Approach

The algorithm we propose fits into the stability-based filtering approach that was first proposed by [DKK+16] (also see the recent survey [DK19]). Using a filtering algorithm is a by-now-standard technique in algorithmic robust statistics, and the approach can be summarized as follows: 1) with high

probability over the sampling of the $n$ uncorrupted samples, there exists a large subset of uncorrupted samples (say, a $1 - O(\epsilon)$ fraction) satisfying a "stability" condition with respect to the mean of the uncorrupted distribution, and 2) a filtering algorithm taking as input a *corrupted* version of the stable samples will remove some of the samples, such that the sample mean of the remaining points is guaranteed to be close to the true mean, which can then be returned as the final estimate. The notion of "stability" depends crucially at the task at hand, and is defined below for the sparse estimation problem.

**Stability-Based Algorithm for Sparsity**    Informally speaking, we say a set $S$ is stable when the mean and the covariance of $S$ do not deviate too much when we remove a few elements from $S$. For the context of sparse estimation, we would like to measure the deviation only along the $k$-sparse directions. However, it is NP-hard to calculate the maximum of $v^\top A v$ over $k$-sparse unit vectors for an arbitrary matrix $A$ (this is known as the sparse PCA problem [TP14]). Following [BDLS17], our definition of stability would involve a convex relaxation of the above optimization problem, using the following definition of the set $\mathcal{X}_k$ and the matrix norm $\| \cdot \|_{\mathcal{X}_k}$.

**Definition 1.4** (The set $\mathcal{X}_k$ and the norm $\| \cdot \|_{\mathcal{X}_k}$)**.** *The set $\mathcal{X}_k$ is defined as the set of positive semi-definite matrices that have trace 1 and $\ell_1$-norm at most $k$ when flattened as a vector. The matrix norm $\|A\|_{\mathcal{X}_k}$ is then defined as $\sup_{M \in \mathcal{X}_k} |A \bullet M|$, where $A \bullet M$ denotes the trace product $\mathrm{tr}(A^\top M)$.*

Note that for any square matrix $A$, $\|A\|_{\mathcal{X}_k}$ is always upper bounded by its spectral norm. Furthermore, observe that for any square matrix $A$, the maximum of $v^\top A v$ over $k$-sparse unit vectors is upper bounded by $\|A\|_{\mathcal{X}_k}$, and the latter can be calculated efficiently using a convex program. We are now ready to define the stability condition.

**Definition 1.5** (Stability Condition)**.** *For $0 < \epsilon < 0.5$ and $\epsilon \leq \delta$, a set $S$ is $(\epsilon, \delta, k)$-stable with respect to $\mu \in \mathbb{R}^d$ and $\sigma \in \mathbb{R}_+$ if it satisfies the following condition: for all subsets $S' \subset S$ with $|S'| \geq (1 - \epsilon)|S|$, the following holds: (i) $\|\mu_{S'} - \mu\|_{2,k} \leq \sigma\delta$, and (ii) $\|\overline{\Sigma}_{S'} - \sigma^2 I\|_{\mathcal{X}_k} \leq \sigma^2 \delta^2/\epsilon$, where $\mu_{S'} = (1/|S'|)\sum_{x \in S'} x$ is the sample mean of $S'$ and $\overline{\Sigma}_{S'} = (1/|S'|)\sum_{x \in S'}(x - \mu)(x - \mu)^\top$ is the second moment of $S'$.*

Definition 1.5 is intended for distributions with covariance matrices at most $\sigma^2$ times identity. We will omit $\mu$ and $\sigma$ above when they are clear from context. Focusing on the Gaussian distribution with identity covariance, [BDLS17] gave a computationally-efficient algorithm with $k^2 \log d$ samples.[1]

By using the standard median-of-means pre-processing described in Section 2, we can reduce the problem to the case when the corruption parameter $\epsilon$ is constant, say $0.01$, and aim to achieve only a constant estimator error in the $\ell_{2,k}$ norm. For this regime, we state the guarantees of robust sparse mean estimation algorithm of [BDLS17] (developed for the Gaussian setting) as follows[2]:

**Fact 1.6.** *Let $S$ be a set in $\mathbb{R}^d$ such that there exists a set $S' \subseteq S$ such that (i) $|S'| \geq 0.99|S|$ and (ii) $S'$ is an $(0.01, O(1), k)$-stable with respect to (unknown) $\mu$ and (unknown) $\sigma$. There is a $\mathrm{poly}(|S|, d)$-time algorithm that takes as input $T$, an $0.01$-corruption of $S$, and returns a mean estimate $\hat{\mu}$ such that $\|\hat{\mu} - \mu\|_{2,k} \leq O(\sigma)$.*

Given the prior algorithmic result, the remaining challenge is to show that, even in the setting of heavy-tailed data, a large subset of the uncorrupted samples satisfies the "stability" condition with high probability. In the dense setting, [DKP20, HLZ20] showed that $O(d)$ samples suffice for stability, which is too large for our setting.

**Truncation is Necessary for Stability**    Recall that our goal is to show that, if we sample $k^2 \log d$ samples from a heavy-tailed distribution, then it contains a large stable subset. For the light-tailed data (Gaussian), this was shown in [BDLS17]. However, this statement is not true for general heavy-tailed distributions. Consider the standard setting for modelling heavy-tailed data, that the covariance $\Sigma$ of the uncorrupted distribution is upper bounded by the identity. For simplicity, also assume that the sparsity parameter $k$, corruption parameter $\epsilon$ and failure probability $\tau$ are all constants. Thus, our goal is to show that, with high probability, there is a large stable subset among $\log d$ samples. Yet, as we show in Example 1 in Section 3, there exists a distribution where deterministically for *any* set of even $o(d)$ many uncorrupted samples, *no* large subset can be stable. This distribution is the one

---

[1]The additional factor of $k$ in their sample complexity (cf. Fact 1.2) is because the convex relaxation involving $\mathcal{X}_k$ norm can be loose. However, [DKS17, BB20] suggest that $k^2$ samples are needed for efficient algorithms.

[2]See also [ZJS21] for a related algorithm.

returning a vector of length $\sqrt{d}$ from a randomly chosen axis direction, which has unit covariance. Essentially, the long length of $\sqrt{d}$ along directions as sparse as the axis directions causes stability to fail to hold.

In order to circumvent this obstacle, we propose to "truncate" all the samples in $\ell_\infty$ norm, before using a stability-based filtering robust mean estimation algorithm: compute an initial mean estimate, then clip each sample coordinate-wise to within a radius of $\Theta(\sqrt{k})$ of the initial mean estimate. This radius is chosen carefully to ensure that the mean of the original distribution and the clipped distribution is close in $\ell_{2,k}$-norm. Ensuring that the clipped distribution also has small variance turns out to be non-trivial as we detail below.

**Necessity of Bounded Higher Moments**  After truncation, no point is too far from the true mean, but truncation can potentially also *rotate* a point about the true mean, in the sense that for a sample, the direction of its difference from the true mean may change after such truncation. In general, this rotation effect can cause much of the mass of the distribution to rotate and concentrate towards certain directions, and significantly *increase* the variance in those directions. (See Appendix A.4 for more details.) In this work, we identify the mild condition that the $4^{\text{th}}$ moment is bounded along each axis direction by some constant, on top of our assumption that $\Sigma \preceq I$, to be sufficient to show that truncation can only increase variance in directions that are non-sparse, in the sense that the resulting covariance will still have bounded $\mathcal{X}_k$ norm (see Lemma 3.1). Thus, under these mild conditions, we can safely truncate our samples (which is necessary for stability to hold as outlined above), and modify our goal to show this truncated distribution contains a large stable set with high probability.

**Stability of Truncated Samples with High Probability**  Even after truncation and after imposing an axis-wise $4^{\text{th}}$ moment bound, it remains challenging to show that, with high probability, there is a large subset of samples that are stable with respect to the true mean.

As we see in Section 5, the analysis reduces to showing that with high probability over the uncorrupted samples, for every matrix $M \in \mathcal{X}_k$, there exists a large subset of samples $S$ whose empirical covariance $\overline{\Sigma}_S$ has a small inner product with $M$, namely that $M \bullet \overline{\Sigma}_S$ is bounded. In the non-sparse setting, the strategy used in [DL22b] and [DKP20] is to first show a high probability event for all $M = vv^\top$ for unit vectors $v$, and then to show that the event for all $M = vv^\top$ deterministically implies that the event holds also for all $M \succeq 0$ with $\text{tr}(M) = 1$. This strategy is important because although the cover of PSD matrices would roughly be exponential in $d^2$, the cover of $vv^\top$ is only exponential in $d$. Thus the first step holds with roughly $d$ samples, and the second step crucially uses the spectral decomposition (SVD) of positive semi-definite (PSD) matrices. On the other hand, in our sparse setting, if we applied the usual SVD to the PSD matrices $M \in \mathcal{X}_k$, the resulting decomposition will generally not yield sparse components, and thus not allowing us to leverage sparsity. Instead, inspired by some matrix norm results derived by Li [Li18], we carefully design a (non-spectral) decomposition that does yield $k^2$-sparse components and can be covered with $k^2 \log d$ samples, as well as a more delicate argument to complete the second step, namely that the event holding for all components $M$ in the decomposition implies the event holding for all $M \in \mathcal{X}_k$. The intricacies of these arguments also allow us to get a sample complexity that ultimately yields an additive (instead of multiplicative) dependence on $\log 1/\tau$, which as described in the previous section is a crucial feature of our result, and in line with the non-robust non-sparse sub-Gaussian mean estimation setting.

### 1.4  Related Work

**Algorithmic Robust Statistics**  The goal of algorithmic robust statistics is to obtain dimension-independent asymptotic error even in the presence of constant fraction of outliers in high dimensions in a computationally efficient way. Since the dissemination of [DKK$^+$16, LRV16], which focused on high-dimensional robust mean estimation, the body of work in the field has grown rapidly. For example, prior work has obtained dimension-independent guarantees for various problems such as linear regression [KKM18, DKS19] and convex optimization [PSBR20, DKK$^+$19a]. See the recent survey [DK19] for a more detailed description. Most relevant to us are the works for robust mean estimation that leverage the sparsity constraints and obtain improved sample complexity. The algorithms developed in [BDLS17, DKK$^+$19b, CDG$^+$, DKK$^+$22] obtain optimal asymptotic error for light-tailed distributions such as Gaussians. However, these algorithms crucially rely on the light-tails and, as outlined in Section 1.3, provably do not work for heavy-tailed distributions.

**Heavy-Tailed Estimation** The recent decades also saw a growing interest in studying statistics in heavy-tailed settings. Even for the basic question of univariate mean estimation without sample corruption, the statistical limits are only recently resolved by a line of work started by Catoni [Cat12] and ending with Lee and Valiant [LV21] (see also [Min22] for an alternative estimator). There is also much research effort on resolving the statistical limits of various high-dimensional estimation tasks for heavy-tailed distributions, for example, mean estimation in $\ell_2$ norm [LM19c] and in other norms [LM19b, DL22a], covariance estimation [MZ20], and stochastic convex optimization [BM22]. In absence of contamination, the goal is to obtain sample complexity as if the distribution were Gaussian. Roughly speaking, this corresponds to an additive dependence on the logarithm of failure probability in various estimation tasks (as we achieve also in this work). We refer the reader to the recent survey for more details [LM19a]. This line of work focuses on the statistical limits, and the estimators developed are generally computationally inefficient.

A closely-related body of research aims to obtain *efficient* algorithms for heavy-tailed distributions with optimal statistical performance, ideally matching the above guarantees. These works include high-dimensional (dense) mean estimation [Hop20, CFB19, DL22b, LLVZ20, DKP20, HLZ20, CTBJ22, LV22], linear regression [CHK+20, PJL20, Dep20b], and covariance estimation [CHK+20]. We note that many of these works are inspired by the algorithmic robust statistics literature, and can also tolerate a constant fraction of contaminated data.

To the best of our knowledge, none of these works studies sparse estimation under heavy-tailed distributions (even in absence of outliers), and our work is the first result with sample complexity that is additive in the logarithm of the failure probability.

## 2 Preliminaries

**Notations** Here we define the notations we use in the rest of the paper. For a (multi-)set $S \subset \mathbb{R}^d$, we denote $\mu_S = (1/|S|) \sum_{x \in S} x$ and $\Sigma_S = (1/|S|)(\sum_{x \in S}(x - \mu_S)(x - \mu_S)^\top)$. When the vector $\mu$ notation is clear from context, we use $\overline{\Sigma}_S$ to denote $(1/|S|) \sum_{x \in S}(x - \mu)(x - \mu)^\top$.

Let $\mathcal{U}_k$ denote the set of $k$-sparse unit vectors in $\mathbb{R}^d$. For two vectors $x$ and $y$, $\langle x, y \rangle$ denotes the dot product $x^\top y$. For a vector $x \in \mathbb{R}^d$, we use $\|x\|_{2,k} := \sup_{v \in \mathcal{U}_k} \langle x, v \rangle$ and $\|x\|_\infty$ to denote $\max_j |x_j|$. For a matrix $M$, we use $\|M\|_1$ to denote $\sum_{i,j} |M_{i,j}|$ and $\|M\|_0$ to denote the number of non-zero entries of $M$. For two matrices $A$ and $B$, we use $A \bullet B$ to denote the trace inner product $\mathrm{tr}(A^\top B)$. Define $\mathcal{X}_k := \{M : M \succeq 0, \mathrm{tr}(M) = 1, \|M\|_1 \leq k\}$. For a matrix $A$, we define $\|A\|_{\mathcal{X}_k} := \sup_{M \in \mathcal{X}_k} |A \bullet M|$. For an $n \in \mathbb{N}$, we use $[n]$ to denote the set $\{1, \ldots, n\}$.

**Coordinate-wise Median-of-Means** We use the coordinate-wise median-of-means algorithm to robustly obtain a preliminary mean estimate, with guarantees captured by the following fact.

**Fact 2.1.** *The coordinate-wise median-of-means algorithm satisfies the following guarantee: given the corruption parameter $\epsilon$, failure probability $\tau$, and a set $T$ of $n$ many $\epsilon$-corrupted samples from a distribution $D$ with mean $\mu$ and axis-wise variance $\mathbb{E}_{X \sim D}[(X_j - \mu_j)^2] \leq \sigma^2$ for all $j \in [d]$, then with probability at least $1 - \tau$ over the sample set $T$, the output of the algorithm $\hat{\mu}$ is such that $\|\hat{\mu} - \mu\|_\infty \leq \sigma O(\sqrt{\epsilon} + \sqrt{(\log(d/\tau))/n})$.*

**Median-of-Means Pre-Processing** Another standard technique we use in this paper is the median-of-means pre-processing, which is a distinct technique from the coordinate-wise median-of-means algorithm mentioned right above. Recall that in Theorem 1.3, the asymptotic error term is $\sqrt{\epsilon}$, which tends to 0 as the corruption parameter $\epsilon \to 0$. The following pre-processing step allows us to reduce the problem from the $\epsilon \to 0$ case to a constant $\epsilon$ case: Split the samples randomly into $g$ equally-sized groups of size $m = n/g$ where $g = \Theta(\epsilon n)$, and replace each group by the sample mean of the group. The effects of this pre-processing is captured by the following Fact 2.2, which we prove for completeness.

**Fact 2.2** (Median-of-Means Pre-Processing). *Suppose there is an efficient algorithm such that, on input $\sigma \in \mathbb{R}_+$ and a 0.1-corrupted set of $n \gg k^2 \log d + \log(1/\tau)$ samples from a distribution $D$ with mean $\mu$ and covariance $\Sigma$ with $\|\Sigma\|_{\mathcal{X}_k} \leq \sigma^2$ and $\mathbb{E}_{X \sim D}[(X_j - \mu_j)^4] = O(\sigma^4)$ for each coordinate $j \in [d]$, returns $\hat{\mu}$ such that $\|\hat{\mu} - \mu\|_{2,k} \leq O(\sigma)$ with probability at least $1 - \tau$.*

*Then, there is an efficient algorithm such that, on input $\epsilon \in (0, 0.1)$ and an $\epsilon$-corrupted set of $n \gg (k^2 \log d + \log(1/\tau))/\epsilon$ samples from a distribution with mean $\mu$ and covariance $\Sigma$, satisfying $\|\Sigma\|_{\mathcal{X}_k} \leq 1$ satisfying $\mathbb{E}_{X \sim D}[(X_j - \mu_j)^4] = O(1)$ for every coordinate $j \in [d]$, returns a mean estimate $\hat{\mu}$ such that $\|\hat{\mu} - \mu\|_{2,k} \leq O(\sqrt{\epsilon})$ with probability at least $1 - \tau$.*

## 3 Truncation Pre-Processing

The general approach for using a stability-based filtering algorithm, for algorithmic robust statistics, is to show that given sufficiently many samples, there exists a large (say $1 - O(\epsilon)$ fraction) subset of the samples that are stable with respect to the true mean $\mu$. However, the following simple example shows that it is not possible for i.i.d. samples from a heavy-tailed distribution to satisfy the sparse stability in $\mathrm{poly}(\log d)$ samples.

**Example 1.** *For any number of moments $t \geq 2$, there is a distribution $X$ satisfying the following conditions: (i) The mean of $X$ is 0, and for every unit vector $v$, the $t^{th}$ moment in direction $v$ is upper bounded by 1, that is, $\mathbb{E}[|\langle v, x \rangle|^t] \leq 1$ for $t \geq 2$, (ii) If $S$ is an arbitrary set of $n \leq o(d^{2/t})$ points from the support of $X$, then the set $S$ cannot be $(\epsilon, O(\sqrt{\epsilon}), k)$-stable, for any $\epsilon > 0$, with respect to the mean of the distribution. As a corollary, no subset of $S$ can be stable either.*

We show Example 1 in Appendix B.

Thus, we need to modify the algorithm if we want to do robust sparse mean estimation using $\mathrm{poly}(k, \log d)$ samples. Our approach is to perform an initial truncation of the samples, before using a stability-based robust mean estimator. A balance needs to be struck, in order to truncate sufficiently for stability to hold (with high probability over the samples), but also to truncate mildly enough such that the mean (and covariance) of the truncated distribution does not shift too much.

For a scalar $a \in \mathbb{R}_+$ and a vector $b \in \mathbb{R}^d$, let $h_{a,b} : \mathbb{R}^d \to \mathbb{R}^d$ be the following thresholding function:

$$\forall i \in [d], \quad h_{a,b}(x)_i = \begin{cases} x_i, & \text{if } |x_i - b_i| \leq a \\ b_i + a & \text{if } x_i - b_i \geq a \\ b_i - a & \text{if } x_i - b_i \leq -a \end{cases}. \tag{1}$$

Note that $h_{a,b}(x)$ projects the point $x$ to the $\ell_\infty$ ball of radius $a$ around $b$.

As explained in the Introduction, truncation in general rotates a point about the true mean, and thus can in fact cause the covariance of the distribution to grow in certain directions. The following lemma captures the fact that, if we make the further mild assumption that the distribution has bounded $4^{\text{th}}$ moment along all the axis directions, then we will at least be able to preserve the $\mathcal{X}_k$ norm of the covariance matrix. The proof of Lemma 3.1 is also in Appendix B.

**Lemma 3.1** (Truncation in $\ell_\infty$). *Let $P$ be a distribution over $\mathbb{R}^d$ with mean $\mu_P$ and covariance $\Sigma_P$, with $\|\Sigma\|_{\mathcal{X}_k} \leq \sigma^2$ for some $\sigma^2 > 0$. Let $X \sim P$ and assume that for all $j \in [d]$, $\mathbb{E}[(X - \mu_P)_j^4] \leq \sigma^4 \nu^4$ for some $\nu \geq 1$. Let $b \in \mathbb{R}^d$ be such that $\|b - \mu\|_\infty \leq a/2$ and $a := 2\sigma\sqrt{k/\epsilon}$ for some $\epsilon \in (0, 1)$. Define $Q$ to be the distribution of $Y := h_{a,b}(X)$. Let the mean and covariance of $Q$ be $\mu_Q$ and $\Sigma_Q$ respectively. Then the following hold:*

*(1) $\|\mu_P - \mu_Q\|_\infty \leq \sigma\sqrt{\epsilon/k}$*

*(2) $\|\mu_P - \mu_Q\|_{2,k} \leq \sigma\sqrt{\epsilon}$*

*(3) $\|\Sigma_P - \Sigma_Q\|_{\mathcal{X}_k} \leq 3\sigma^2\epsilon\nu^4$*

*(4) For all $i \in [d]$, $\mathbb{E}[(Y - \mu_Q)_i^4] \leq 8\nu^4\sigma^4$*

*(5) $\|Y - \mu_Q\|_\infty \leq 2a = 4\sigma\sqrt{k/\epsilon}$ almost surely.*

In Lemma 3.1 above, $b$ represents the initial mean estimate, and $\tilde{\mu}$ will be obtained by Fact 2.1.

## 4 Algorithm and Analysis

The high-level algorithm we propose is stated as follows.

---
**Algorithm 1** Robust Sparse Mean Estimation with High Probability
---

1. Input: An $\epsilon$-corrupted sample set $S \subseteq \mathbb{R}^d$ of size $n$

2. Median-of-Means pre-processing: Group points into $g$ groups, each of size $m = n/g$, where $g = \Theta(\epsilon n)$, and take the sample mean of a group to be a new point

3. Compute initial coordinate-wise median-of-means estimate $\tilde{\mu}$

4. Truncate all points to within $B_\infty(\tilde{\mu}, 2\sqrt{k})$, namely, given a point $x$, we replace it with the point $h_{2\sqrt{k},\tilde{\mu}}(x)$, where $h_{a,b}$ is defined in Equation (1).

5. Run the stability-based robust *sparse* mean estimator from Fact 1.6 on the samples after the processing of Step 4.

---

We note that this algorithm is shift and scale invariant, based on the same invariance of the median-of-means pre-processing as well as the invariance of the robust sparse mean estimator from Fact 1.6.

We will now prove that Algorithm 1 satisfies the guarantees of Theorem 1.3 and its stronger version, Theorem D.12. Our analysis uses Theorem D.11, which states that, with high probability, there exists a set consisting of most of the truncated samples that is stable with respect to some point close to the true mean in $\ell_\infty$ norm. In Section 5, we outline the central part (Theorem 5.1) of the proof of this theorem. We have not optimized constants in our analysis.

*Proof of Theorems 1.3 and D.12.* By Fact 2.2, it suffices to show that, for every $\sigma > 0$, Steps 3–5 in Algorithm 1 yields an $O(\sigma)$ estimation error in $\ell_{2,k}$ norm when given corrupted samples from a distribution $D$ with covariance bounded by $\sigma^2 I$ and axis-wise 4th moment bounded by $O(\sigma^4)$.

Theorem D.11 states that, with probability at least $1 - \tau$, the samples after the processing of Step 4 are such that there exists a 95% of the samples that form a $(0.1, O(1), k)$-stable subset with respect to some vector $\mu'$ and $\sigma$ with $\|\mu' - \mu\|_\infty \leq O(\sigma/\sqrt{k})$. Fact 1.6 then guarantees that, on input such a set of samples, the algorithm we invoke Step 5 of Algorithm 1 will return a mean estimate $\hat{\mu}$ such that $\|\hat{\mu} - \mu'\|_{2,k} \leq O(\sigma)$. Further, since $\|\mu' - \mu\|_\infty \leq O(\sigma/\sqrt{k})$, we have that $\|\mu' - \mu\|_{2,k} \leq O(\sigma)$, and therefore we can conclude with the triangle inequality that the mean estimate $\hat{\mu}$ satisfies $\|\hat{\mu} - \mu\|_{2,k} \leq O(\sigma)$. $\qquad\square$

# 5 Stability After Removing Points: Additive dependence on $\log(1/\tau)$

In this section, we sketch the main stability result in this paper. Recall, via the median-of-means pre-processing, that we only need to consider the constant contamination case ($\epsilon = \Theta(1)$). Thus, the goal is to show (Theorem D.11) that with high probability, after truncation according to the coordinate-wise median-of-means preliminary estimate, there exists a large subset of uncontaminated samples that is $(\Theta(1), O(\sigma), k)$-stable with respect to the *true mean* of the distribution where $\sigma = \|\Sigma\|_{\mathcal{X}_k}$.

Theorem 5.1 below, an informal version of Theorem C.1 in Appendix C, captures the core of the argument. The key difference between Theorems 5.1 and D.11 is that the former is a stability result concerning uncontaminated samples truncated according to some *fixed* vector close to the true mean. On the other hand, the final stability result we require concerns samples truncated according to the coordinate-wise median-of-means estimate, which itself depends on the samples and is not fixed. Appendix D shows the delicate argument going from Theorem 5.1 to Theorem D.11.

**Theorem 5.1** (Informal version of Theorem C.1). *Let $S$ be a set of $n$ i.i.d. data points from a distribution $P$ over $\mathbb{R}^d$. Let the mean of $P$ be $\mu$, and covariance $\Sigma$ such that $\|\Sigma\|_{\mathcal{X}_k} \leq \sigma^2$, and for all $j \in [d]$, $\mathbb{E}[X_j^4] = O(\sigma^4)$. Suppose $P$ is supported over the set $\{x : \|x - \mu\|_\infty = O(\sigma\sqrt{k})\}$. If $n = \Omega(k^2 \log d + \log(1/\tau))$, then, with probability $1 - \tau$, there exists a set $S' \subset S$ such that:*

1. *$|S'| \geq 0.98n$*

2. *$S'$ is $(0.01, \delta, k)$-stable with respect to $\mu$ and $\sigma$ where $\delta = O(1)$.*

For the rest of the section, we sketch the proof of Theorem 5.1.

*Proof sketch for Theorem 5.1.* In the following proof sketch, we hide most multiplicative constants. The proof of Theorem C.1 in Appendix C will make explicit the calculations for completeness. For the rest of the sketch, we will also assume $\mu = 0$ without loss of generality.

Instead of directly showing the existence of subset $S' \subseteq S$ (with high probability over the samples $S$) that is stable, we will be working with weights/distributions over the set $S$ of samples. Given a weight vector $w$ and $S = \{x_1, \ldots, x_n\} \subset \mathbb{R}^d$, denote $\overline{\Sigma}_w$ by $\sum_i w_i x_i x_i^\top$. Then, Proposition A.1 in Appendix A lets us show the following simpler condition instead: let $\Delta_{n,\epsilon}$ be the set of weights/distributions $w$ such that $w_i \leq 1/(1 - \epsilon)$, then there exists a weighting $w \in \Delta_{n,0.01}$ such that $\|\overline{\Sigma}_w\|_{\mathcal{X}_k} \leq O(\sigma^2)$. That is, for the following proof, we just need to prove that $\min_{w \in \Delta_{n,0.01}} \|\overline{\Sigma}_w\|_{\mathcal{X}_k} = O(\sigma^2)$.

We proceed as follows:

$$\min_{w \in \Delta_{n,0.01}} \|\overline{\Sigma}_w\|_{\mathcal{X}_k} = \min_{w \in \Delta_{n,0.01}} \max_{M \in \mathcal{X}_k} M \bullet \overline{\Sigma}_w = \max_{M \in \mathcal{X}_k} \min_{w_M \in \Delta_{n,0.01}} M \bullet \overline{\Sigma}_w$$

where the last equality is a straightforward application of the minimax theorem for a minimax optimization problem with independent convex domains and a bilinear objective. It thus suffices to show the following: with probability $1 - \tau$,

$$\forall M \in \mathcal{X}_k : \ |\{x \in S : x_i^\top M x_i \gg \sigma^2\}| \leq 0.01 |S| \tag{2}$$

and we can construct the weight $w_M$ as uniform distribution over the elements outside the above set.

Define the following sets of sparse matrices, the first of which also appears in [Li18]:

$$\mathcal{A}_k := \left\{ A \in \mathbb{R}^{d \times d} : \|A\|_0 \leq k^2, \|A\|_F \leq 1 \right\},$$
$$\mathcal{A}_{k,P} := \left\{ A \in \mathcal{A}_k : \mathbb{P}_{x \sim P} \left( x^\top A x \gg \sigma^2 \right) \leq \text{small constant} \right\}. \tag{3}$$

If $n \gg k^2 \log d + \log(1/\tau)$, then a standard covering/VC-dimension bound (see Lemma B.2 for details) implies that the following event holds with probability $1 - \tau$:

$$\forall A \in \mathcal{A}_{k,P} : \ |\{x \in S : x^\top A x \gg \sigma^2\}| \leq 0.002 \cdot |S|. \tag{4}$$

We will now show that the event in Equation (4) implies that the event in Equation (2) holds. Note that the constants hidden in the $\gg$ notation in Equations Equation (2) and Equation (4) are different, which is the slackness that makes the implication possible. As mentioned, we will make these constants explicit in the formal proof in Appendix C.

Suppose, for the sake of contradiction, that the event in Equation (2) does not hold. Then there exists an $M \in \mathcal{X}_k$ such that $|\{x \in S : x_i^\top M x_i \gg \sigma^2\}| > 0.01 |S|$. We will show the existence of a matrix $Q$ violating event Equation (4), via the probabilistic method, to reach the desired contradiction.

To achieve this, we define a *decomposition* of $M \in \mathcal{X}_k$ into components in $\mathcal{A}_{k,P}$, in the sense of a distribution over matrices $Q$ such that 1) $\mathbb{E}[Q] = M$ and 2) a small constant multiple of $Q$ belongs to $\mathcal{A}_{k,P}$ with high constant probability. We will additionally show that this decomposition $Q$ roughly preserves quadratic forms, in that if $M$ violates Equation (2) then $Q$ violates Equation (4) with non-trivial probability.

Fixing an $M$ that violates Equation (2), consider the random matrix $Q$ where each entry $Q_{i,j}$ is independently set to be $M_{i,j}/p_{i,j}$ with probability $p_{i,j}$ and 0 otherwise, where $p_{i,j} := \min(1, k|M_{i,j}|/c)$ for a large constant $c > 0$.

We will show that the following events hold simultaneously with non-zero probability, leading to a contradiction to event Equation (4): (i) $Q/c' \in \mathcal{A}_{k,P}$ for a large constant $c'$ and (ii) $|\{x \in S : x_i^\top (Q/c') x_i \gg \sigma^2\}| > 0.002 \cdot |S|$. Using different techniques, we will show that the first condition holds with probability at least $1 - 2 \times 10^{-6}$ and the second condition holds with probability at least $4 \times 10^{-6}$, thus implying that the events hold simultaneously with non-zero probability.

**Condition (i), that $Q/c' \in \mathcal{A}_{k,P}$ with high constant probability.** Showing that $Q/c'$ belongs to $\mathcal{A}_k$ is straightforward: by the construction of $Q$, it has small expected sparsity as well as small expected Frobenius norm. An application of Markov's inequality shows that $Q/c' \in \mathcal{A}_k$ with high constant probability (Lemma C.2).

The trickier part is to show that $Q/c'$ is also in $\mathcal{A}_{k,P}$, namely that $\mathbb{P}_{x \sim P}(x^\top(Q/c')x \gg \sigma^2)$ is upper bounded by the small constant. We consider the distribution of $x^\top Q x$ over the probability of independently drawing $x \sim P$ and a random $Q$, and show that $x^\top Q x$ is small with high probability over this joint distribution (Lemma C.4), which requires using the axis-wise $4^{\text{th}}$ moment bounds on $P$ as well as the fact that $M \in \mathcal{X}_k$. Lemma C.3 implies that with high probability, we will draw a $Q$ satisfying $\mathbb{P}_{x \sim P}(x^\top Q x \gg \sigma^2)$ being bounded by a small constant.

**Condition (ii), that if $M$ violates Equation (2) then $Q$ violates Equation (4) with non-trivial probability** We consider the random variable $Z = \sum_i \mathbb{1}_{x_i^\top Q x_i \gg \sigma^2}$, over the randomness of $Q$. The overall strategy is to lower bound $\mathbb{E}[Z] \geq cn$ for a small $c > 0$ and use the Paley-Zygmund inequality to show that with at least small constant probability, $Z \geq cn/2$, where we use $Z$ always is less than $n$. The expectation of $Z$ is just the average over all samples the probability that $x_i^\top Q z_i \gg \sigma^2$. Since the fraction of $x_i$s with $x_i^\top M x_i \gg \sigma^2$ is at least a constant, to lower bound $\mathbb{E}[Z]$, it suffices for us to lower bound this probability for each such $x_i$. Fix any such $x$ for the remainder of the argument.

Lower bounding the probability that $x^\top Q x \gg \sigma^2$ is non-trivial, since, the support of the quadratic form can scale with $k$ and given a lower bound on its expectation, it could be the case that it is 0 with high probability and has a huge value with small probability.

We will argue that this does not happen by considering two cases. First, and the simpler setting, is when $x^\top Q x$ has small variance. In this case, we can use Chebyshev's inequality to lower bound the probability that $x^\top Q x \gg \sigma^2$. Essentially, this is the case when $x^\top Q x$ is "concentrated".

The second, and the more involved setting, is when $x^\top Q x$ has large variance. Here, we would like to argue that $x^\top Q x$ is sufficiently "anti-concentrated" like a Gaussian, in the sense that there is a constant probability mass above its mean. Noting that $x^\top Q x = \sum_{i \in [d], j \in [d]}(x)_i(x)_j Q_{i,j}$ and the entries of $Q$ are independent, this is a sum of independent terms, and we can reasonably expect Gaussian-like behavior. Concretely, we will show this using the Berry-Esseen theorem, which requires us to upper bound the third moment of $x^\top Q x$. This upper bound crucially uses the facts that 1) the samples here are truncated, which implies that each $(x)_i(x)_j$ has bounded magnitude and 2) each $Q_{i,j}$ is a (scaled) Bernoulli which has exact and simple expressions for its second and third moments.

Summarizing, we argued that conditions (i) and (ii) hold simultaneously with non-zero probability, thus showing the existence of a $Q$ violating Equation (4), reaching a contradiction assuming the existence of some $M$ violating Equation (2). Therefore, Equation (4) implies Equation (2), which completes the argument for Theorem 5.1. □

## Acknowledgements

We thank the anonymous reviewers for helpful comments on the paper. Ilias Diakonikolas was supported by NSF Medium Award CCF-2107079, NSF Award CCF-1652862 (CAREER), a Sloan Research Fellowship, and a DARPA Learning with Less Labels (LwLL) grant. Daniel Kane was supported by NSF Medium Award CCF-2107547, NSF Award CCF-1553288 (CAREER), and a grant from CasperLabs. Jasper C.H. Lee was supported in part by the generous funding of a Croucher Fellowship for Postdoctoral Research, NSF award DMS-2023239, NSF Medium Award CCF-2107079 and NSF AiTF Award CCF-2006206. Ankit Pensia was supported by NSF grants NSF Award CCF-1652862 (CAREER), DMS-1749857, and CCF-1841190.

Part of this work was done while Ilias Diakonikolas, Jasper C.H. Lee and Ankit Pensia were visitors at the Simons Institute for the Theory of Computing.

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
