# Supplementary Material

**Additional Notation.** For a vector $x \in \mathbb{R}^d$ and $H \subset [d]$, we denote $v_H$ to denote the vector that is equal to $v$ on $i \in H$, and zero otherwise. For a real-valued random variable $X$ and $m \in \mathbb{N}$, we use $\|X\|_{L_m}$ to denote $(\mathbb{E}\,|X|^m)^{1/m}$. For a set $S \subseteq \mathbb{R}^d$ and a function $f$, we also define the set function notation $f(S)$ as $\{f(x) \,|\, x \in S\}$.

## A   Miscellaneous Lemmas and Facts

### A.1   Finding a stable subset from a stable weighted subset

For a set $S$ on $n$ points, we define $\Delta_{n,\epsilon}$ as the set of weights $w \in \mathbb{R}^n$ such that $w_i \in [0, 1/((1-\epsilon)n]$ for all $i \in [n]$ and $\sum_i w_i = 1$. For a fixed vector $\mu \in \mathbb{R}^d$ that will be clear from context, a set of $n$ points $S = \{x_1, \ldots, x_n\}$, and weights $w \in \Delta_{n,\epsilon}$ over $S$, we use $\overline{\Sigma}_w$ to denote $\sum_i w_i(x_i - \mu)(x_i - \mu)^\top$.

The goal of this section is to show Proposition A.1, which states that if we have a weight $w$ over $S$ such that $\overline{\Sigma}_w$ (with respect to some vector $\mu$) has bounded $\mathcal{X}_k$ norm proportional to $\sigma^2$ for some $\sigma > 0$, then there must exists some large subset $S' \subseteq S$ that is stable with respect to $\mu$ and $\sigma$.

**Proposition A.1.** *Let $S$ be a set of $n$ points in $\mathbb{R}^d$. Let $\Delta_{n,\epsilon}$ be the set of weights defined above, and define the notation $\overline{\Sigma}_w = \sum_{x_i \in S} w_i(x_i - \mu)(x_i - \mu)^\top$ for some given vector $\mu \in \mathbb{R}^d$. Suppose that there exists a $w \in \Delta_{n,\epsilon}$ such that $\|\overline{\Sigma}_w\|_{\mathcal{X}_k} \leq B\sigma^2$ for some vector $\mu$. Then there exists a subset $S' \subseteq S$ such that $(i)\,|S'| \geq (1 - 2\epsilon)n$ and $(ii)$ $S'$ is $(\epsilon, \delta, k)$-stable with respect to $\mu$ and $\sigma$, where $\delta = O(\sqrt{B} + 1)$.*

Observe that $\|\overline{\Sigma}_w\|_{\mathcal{X}_k} \leq B\sigma^2$ implies $\|\overline{\Sigma}_w - \sigma^2 I\|_{\mathcal{X}_k} \leq (B + 1)\sigma^2$ by the triangle inequality. In order to show Proposition A.1, we show Lemma A.2, which is a weakening of Proposition A.1 where we additionally assume that $\mu_w = \sum_i w_i x_i$ is close to $\mu$, where $\mu$ is the vector we use to define $\overline{\Sigma}_w$ as well as the vector that we want to find a large sample subset $S'$ to be stable with respect to. To use Lemma A.2, we additionally show Proposition A.4, which states that $\|\overline{\Sigma}_w\|_{\mathcal{X}_k} \leq B\sigma^2$ is enough to imply that $\mu_w$ is close to $\mu$. We combine Lemma A.2 and Proposition A.4 to prove Proposition A.1 at the end of Appendix A.1.

**Lemma A.2.** *Suppose, for some $\epsilon \leq \frac{1}{3}$ and for some $\delta \geq \sqrt{\epsilon}$, there exist a $w \in \Delta_{n,\epsilon}$ over a set of $n$ samples $S = \{x_1, \ldots, x_n\}$, a $\mu \in \mathbb{R}^d$ and a $\sigma > 0$ such that*

- $\|\mu_w - \mu\|_{2,k} \leq \delta\sigma$,

- $\|\sum_{i \in [n]} w_i(x_i - \mu)(x_i - \mu)^\top - \sigma^2 I\|_{\mathcal{X}_k} \leq \sigma^2 \frac{\delta^2}{\epsilon}$.

*Then, there exists a subset $S' \subseteq S$ of samples such that*

- $|S'| \geq (1 - 2\epsilon)|S|$,

- $S'$ is $(\epsilon, \delta', k)$-stable with respect to $\mu$ and $\sigma$, where $\delta' = O(\delta + \sqrt{\epsilon})$.

*Proof.* Without loss of generality, we will only handle the $\sigma = 1$ case to simplify notation.

The main step is to show the existence of a large subset $S'$ whose mean is within $10\delta + 10\sqrt{\epsilon}$ of $\mu$ and whose variance is at most $9(1 + \delta^2/\epsilon)$. In fact, we can simply choose $S'$ to be the subset whose weights $w_i$ are the largest.

Without loss of generality, assume $\mu = 0$ and that $\epsilon n$ is an integer. We also order the samples in decreasing order of weight in $w$, namely, $1/((1 - \epsilon)n) \geq w_1 \geq w_2 \geq \ldots \geq w_n$.

First, we will lower bound each $w_i$. We have that for each $k \in [n]$,

$$1 = \sum_i w_i \leq \frac{k}{(1-\epsilon)n} + (n - k)w_k,$$

which upon rearranging implies that

$$w_k \geq \frac{(1-\epsilon)n - k}{(1-\epsilon)n(n-k)}.$$

In particular, for $k = (1 - 2\epsilon)n$, we have

$$w_{(1-2\epsilon)n} \geq \frac{1}{2(1-\epsilon)n}.$$

Letting $S'$ to be the $(1 - 2\epsilon)n$ points with largest weight, we have that for all $i \in S'$, $w_i \geq \frac{1}{2(1-\epsilon)n}$. We will use this to now bound the $\mathcal{X}_k$ norm of $\Sigma_{S'} = \frac{1}{|S'|} \sum_{i \in S'} x_i x_i^\top$. Consider an arbitrary $M \in \mathcal{X}_k$, we have

$$
\begin{aligned}
\sum_{i \in S'} \frac{1}{|S'|} \langle x_i x_i^\top, M \rangle &= \sum_{i \in S'} \frac{1}{(1-2\epsilon)n} \langle x_i x_i^\top, M \rangle \\
&\leq \sum_{i \in S'} \frac{2(1-\epsilon)}{1-2\epsilon} w_i \langle x_i x_i^\top, M \rangle \\
&\leq \sum_{i \in S} \frac{2(1-\epsilon)}{1-2\epsilon} w_i \langle x_i x_i^\top, M \rangle \\
&\leq 9\left(1 + \frac{\delta^2}{\epsilon}\right).
\end{aligned}
$$

Since $\delta \geq \sqrt{\epsilon}$, this in turn implies the (rather loose in constants) inequality that $\|\Sigma_{S'} - I\|_{\mathcal{X}_k} \leq 20(\delta^2/\epsilon)$.

Next, we show that the mean $\mu_{S'}$ of $S'$ is $10\delta + 10\sqrt{\epsilon}$-close to $\mu = 0$. This will essentially follow from 1) the uniform distribution $U_{S'}$ over $S'$ is close in total variation distance to $w$ and 2) the contribution of the tail to the mean of a bounded-covariance distribution is small.

For 1), using the notation that $U_S$ is the uniform distribution over $S$ (analogous to the $S'$ notation just before), it is immediate that by the triangle inequality,

$$d_{\mathrm{TV}}(w, U_{S'}) \leq d_{\mathrm{TV}}(w, U_S) + d_{\mathrm{TV}}(U_S, U_{S'}) \leq \epsilon + 2\epsilon = 3\epsilon.$$

A standard consequence is that there exists distributions $p^{(1)}$, $p^{(2)}$ and $p^{(3)}$ such that

$$w = (1 - 3\epsilon)p^{(1)} + 3\epsilon p^{(2)} \quad \text{and} \quad U_{S'} = (1 - 3\epsilon)p^{(1)} + 3\epsilon p^{(3)}.$$

Intuitively, treating $p^{(2)}$ and $p^{(3)}$ as the "tails", we will bound their contributions to the mean under the boundedness of the covariance of $w$ and $U_{S'}$.

Take any $k$-sparse unit vector direction $v \in \mathcal{U}_k$, we can bound the following variances in the direction of $v$:

$$3\epsilon \sum_i p_i^{(2)} \langle x_i, v \rangle^2 \leq \sum_i w_i \langle x_i, v \rangle^2 \leq 1 + \frac{\delta^2}{\epsilon},$$

$$3\epsilon \sum_i p_i^{(3)} \langle x_i, v \rangle^2 \leq \sum_i U_{S',i} \langle x_i, v \rangle^2 \leq 9\left(1 + \frac{\delta^2}{\epsilon}\right),$$

where we used the fact that $vv^\top$ is in $\mathcal{X}_k$ for a $k$-sparse unit vector $v$.

By Jensen's inequality, we can then conclude that

$$\left| 3\epsilon \sum_i p_i^{(2)} \langle x_i, v \rangle \right| \leq \sqrt{3\epsilon} \sqrt{3\epsilon \sum_i p_i^{(2)} \langle x_i, v \rangle^2} \leq \sqrt{3\epsilon} \sqrt{1 + \frac{\delta^2}{\epsilon}} \leq \sqrt{3}(\sqrt{\epsilon} + \delta),$$

$$\left| 3\epsilon \sum_i p_i^{(3)} \langle x_i, v \rangle \right| \leq \sqrt{3\epsilon} \sqrt{3\epsilon \sum_i p_i^{(3)} \langle x_i, v \rangle^2} \leq 3\sqrt{3\epsilon} \sqrt{1 + \frac{\delta^2}{\epsilon}} \leq 3\sqrt{3}(\sqrt{\epsilon} + \delta).$$

Finally, since $U_{S'} = w - 3\epsilon p^{(2)} + 3\epsilon p^{(3)}$, by the triangle inequality, we have

$$
\begin{aligned}
|\langle \mu_{S'} - \mu, v \rangle| &= \left| \sum_i U_{S',i} \langle x_i, v \rangle \right| \\
&\leq \left| \sum_i w_i \langle x_i, v \rangle \right| + \left| 3\epsilon \sum_i p_i^{(2)} \langle x_i, v \rangle \right| + \left| 3\epsilon \sum_i p_i^{(3)} \langle x_i, v \rangle \right| \\
&\leq \delta + \sqrt{3}(\sqrt{\epsilon} + \delta) + 3\sqrt{3}(\sqrt{\epsilon} + \delta) \\
&\leq 10\delta + 10\sqrt{\epsilon},
\end{aligned}
$$

where the second inequality uses the above bounds as well as the assumption that $\|\mu_w - \mu\|_{2,k} \leq \delta$.

Now that we have shown that $\mu_{S'}$ is close to $\mu$ in $2, k$ norm and $\Sigma_{S'}$ is small in the $\mathcal{X}_k$ norm, we will use the following lemma (Lemma A.3) to show that the set $S'$ is $(\epsilon, O(\delta + \sqrt{\epsilon})$-stable with respect to $\mu$. $\qquad \square$

**Lemma A.3** (Bounded Mean and Covariance implies $O(\sqrt{\epsilon})$ stability). *Let $\mu \in \mathbb{R}^d$ and let $S'$ be a set of samples such that $\|\mu_{S'} - \mu\|_{2,k} \leq \delta$ and $\left\| \frac{1}{|S'|} \sum_{x \in S'} (x - \mu)(x - \mu)^\top - I \right\|_{\mathcal{X}_k} \leq \frac{\delta^2}{\epsilon}$ for some $0 \leq \epsilon \leq \delta$ and $\epsilon \leq 0.5$. Then $S'$ is $(\epsilon, \delta', k)$-stable with respect to $\mu$ where $\delta' = O(\delta + \sqrt{\epsilon})$ and $\delta' \geq \sqrt{\epsilon}$.*

*Proof.* Consider an arbitrary large subset $S'' \subseteq S'$ where $|S''| \geq (1 - \epsilon)|S'|$. Without loss of generality, take $\mu = 0$. Then, for an arbitrary $M \in \mathcal{X}_k$,

$$
\langle \overline{\Sigma}_{S''} - I, M \rangle = \frac{1}{S''} \sum_{i \in S''} \langle x_i x_i^\top, M \rangle - 1,
$$

which is trivially at least $-1 \geq -(\delta'^2)/\epsilon$ for $\delta' \geq \sqrt{\epsilon}$. As for the upper bound, we have

$$
\begin{aligned}
\langle \overline{\Sigma} - I, M \rangle &= \frac{1}{S''} \sum_{i \in S''} \langle x_i x_i^\top, M \rangle - 1 \\
&\leq \left( \frac{1}{S''} \sum_{i \in S'} \langle x_i x_i^\top, M \rangle \right) - 1 \\
&\leq \frac{1}{1 - \epsilon} \left( 1 + \frac{\delta^2}{\epsilon} \right) - 1 \\
&= \frac{\frac{\delta^2}{\epsilon} + \epsilon}{1 - \epsilon} \\
&\leq \frac{2}{\epsilon}(\delta^2 + \epsilon^2) \\
&\leq \frac{\delta'^2}{\epsilon},
\end{aligned}
$$

for some $\delta' = \Theta(\delta + \sqrt{\epsilon})$.

We now bound the error in the mean of $S''$ in $2, k$ norm. First, observe that, for an arbitrary $k$-sparse unit vector $v$,

$$\left| \frac{1}{|S'|} \sum_{i \in S' \setminus S''} \langle x_i, v \rangle \right| = \left| \frac{1}{|S'|} \sum_{i \in S'} \mathbb{1}[x_i \in S' \setminus S''] \langle x_i, v \rangle \right|$$

$$\leq \frac{1}{|S'|} \sum_{i \in S'} |\mathbb{1}[x_i \in S' \setminus S''] \langle x_i, v \rangle|$$

$$\leq \sqrt{\epsilon} \sqrt{\frac{1}{|S'|} \sum_{i \in S'} \langle x_i, v \rangle^2}$$

$$\leq \sqrt{\epsilon} \sqrt{1 + \frac{\delta^2}{\epsilon}}$$

$$= \sqrt{\epsilon + \delta^2} \,,$$

where the second inequality is an application of Hölder's inequality, and the third inequality uses the fact that for a unit $k$-sparse vector $v$, $vv^\top$ is in $\mathcal{X}_k$.

Thus, again for an arbitrary $k$-sparse unit vector $v$,

$$|\langle \mu_{S''}, v \rangle| = \left| \frac{1}{|S''|} \sum_{i \in S''} \langle x_i, v \rangle \right|$$

$$\leq \frac{1}{1 - \epsilon} \left| \frac{1}{|S'|} \sum_{i \in S''} \langle x_i, v \rangle \right|$$

$$\leq 2 \left( \left| \frac{1}{|S'|} \sum_{i \in S'} \langle x_i, v \rangle \right| + \left| \frac{1}{|S'|} \sum_{i \in S' \setminus S''} \langle x_i, v \rangle \right| \right)$$

$$\leq 2(\delta + \sqrt{\epsilon + \delta^2}) = O(\delta + \sqrt{\epsilon}) = \delta'.$$

$\square$

**Proposition A.4** (Bounded Covariance and Stability). *Let $\mu \in \mathbb{R}^d$ and let $S$ be a set of $n$ samples. Let $w \in \Delta_{n,\epsilon}$ over the set of samples $S$ such that $\|\sum_i w_i(x_i - \mu)(x_i - \mu)^\top\|_{\mathcal{X}_k} \leq r$ for some $r > 0$. Then $\|\mu_w - \mu\|_{2,k} \leq \sqrt{r}$.*

*Proof.* For every $k$-sparse unit vector $v$, $vv^\top$ is in $\mathcal{X}_k$, and thus for every sparse unit vector $v$, we have that $\sum_i w_i \langle x_i - \mu, v \rangle^2 \leq r$. Applying Cauchy-Schwarz inequality, we get that for any sparse unit vector $v$, it follows that $\sum_i w_i \langle x_i - \mu, v \rangle \leq \sqrt{\sum_i w_i \langle x_i - \mu, v \rangle^2} \leq \sqrt{r}$. $\square$

With Proposition A.4 and Lemma A.2, we can prove Proposition A.1.

*Proof of Proposition A.1.* Without loss of generality, we will assume that $\sigma = 1$. By Proposition A.4, we have that $\|\mu_w - \mu\|_{2,k} \leq \sqrt{B}$. We thus have a weighting $w \in \Delta_{n,\epsilon}$, where $\|\mu_w - \mu\|_{2,k} \leq \delta_0$ and $\|\overline{\Sigma}_w - I\|_{\mathcal{X}_k} \leq \delta_0^2/\epsilon$ for $\delta_0 = \sqrt{B} + 1$, where we use triangle inequality on the $\|\cdot\|_{\mathcal{X}_k}$ norm. By Lemma A.2, we know that there exists a set $S'$ such that $|S'| \geq (1 - 2\epsilon)n$ and $S'$ is $(\epsilon, \delta, k)$-stable with respect to $\mu$ and $\sigma$, where $\delta = O(\delta_0 + \sqrt{\epsilon}) = O(\sqrt{\epsilon} + \sqrt{B} + 1) = O(\sqrt{B} + 1)$. $\square$

## A.2 Median of Means

**Fact 2.2** (Median-of-Means Pre-Processing). *Suppose there is an efficient algorithm such that, on input $\sigma \in \mathbb{R}_+$ and a 0.1-corrupted set of $n \gg k^2 \log d + \log(1/\tau)$ samples from a distribution $D$ with mean $\mu$ and covariance $\Sigma$ with $\|\Sigma\|_{\mathcal{X}_k} \leq \sigma^2$ and $\mathbb{E}_{X \sim D}[(X_j - \mu_j)^4] = O(\sigma^4)$ for each coordinate $j \in [d]$, returns $\hat{\mu}$ such that $\|\hat{\mu} - \mu\|_{2,k} \leq O(\sigma)$ with probability at least $1 - \tau$.*

*Then, there is an efficient algorithm such that, on input $\epsilon \in (0, 0.1)$ and an $\epsilon$-corrupted set of $n \gg (k^2 \log d + \log(1/\tau))/\epsilon$ samples from a distribution with mean $\mu$ and covariance $\Sigma$, satisfying*

$\|\Sigma\|_{\mathcal{X}_k} \leq 1$ *satisfying* $\mathbb{E}_{X \sim D}[(X_j - \mu_j)^4] = O(1)$ *for every coordinate* $j \in [d]$, *returns a mean estimate* $\hat{\mu}$ *such that* $\|\hat{\mu} - \mu\|_{2,k} \leq O(\sqrt{\epsilon})$ *with probability at least* $1 - \tau$.

*Proof.* The new algorithm simply performs median-of-means preprocessing as defined in Section 2 before the fact statement, yielding $g$ new samples that are fed into the algorithm that works with constant corruption. The uncorrupted new samples, namely the ones that are the sample mean of groups containing no originally corrupted samples, are distributed i.i.d. according to the distribution $D'$ which has mean $\mu$, and covariance $\Sigma' = (g/n)\Sigma$, with axis-wise fourth moment $\mathbb{E}_{Y \sim D'}[(Y_j - \mu_j)^4]$ being bounded by $C(g^2/n^2)\,\mathbb{E}_{X \sim D}[(X_j - \mu_j)^4]$ for every $j \in [d]$ for some constant $C > 0$, obtained by the following fact:

**Fact A.5.** *(Marcinkiewicz-Zygmund inequality) Recall the notation* $\|X\|_{L_s}$ *for a centered random variable* $X$, *defined as* $\mathbb{E}[|X|^s]^{1/s}$. *Let* $W_1, \ldots, W_m, W$ *be identical and independent centered random variables on* $\mathbb{R}$ *with a finite* $\|W\|_{L_s}$ *norm for* $s \geq 2$. *Then,*

$$\left\| \frac{1}{m} \sum_{i=1}^{m} W_i \right\|_{L_s} \leq \frac{3\sqrt{s}}{\sqrt{m}} \|W\|_{L_s}.$$

First note that we give $g$ samples to the original algorithm, and $g = \Omega(\epsilon n) = \Omega(k^2 \log d + \log(1/\tau))$ by definition. Next, we need to check that the *normalized* axis-wise 4th moment of $D'$ is $O(1)$ times the (bound on the) $\mathcal{X}_k$-norm of the covariance matrix, that is, for all $j \in [d]$, it holds that $(\mathbb{E}_{X \sim D'}[(X_j - \mu_j)^4])^{1/4} \leq O(\sigma^4)$ and $\|\Sigma'\|_{\mathcal{X}_k} = O(\sigma^2)$. By the calculations at the end of the previous paragraph and the assumptions in the statement, we note that this is true for $\sigma = O(\sqrt{g/n})$.

Lastly, we check that, by the scale-invariance of the original algorithm that works with constant corruption, the estimation error of the final algorithm is upper bounded by $O(\sigma\|\Sigma\|_{\mathcal{X}_k}) = O(\sqrt{(g/n)\|\Sigma\|_{\mathcal{X}_k}}) = O(\sqrt{g/n}) = O(\sqrt{\epsilon})$ as desired. $\qquad\square$

## A.3  $\mathcal{X}_k$-Norm

**Lemma A.6.** *Let* $A \in \mathbb{R}^{d \times d}$ *be a symmetric matrix such that* $|A_{i,i}| \leq \eta_1$ *for each* $i \in [d]$, *and* $|A_{i,j}| \leq \eta_2$ *for each* $i \neq j \in [d] \times [d]$. *Then* $\|A\|_{\mathcal{X}_k} \leq \eta_1 + k\eta_2$.

*Proof.* Let $A = B + C$, where $B$ is a diagonal matrix and $C$ is diagonal-free. Then we have the following using triangle inequality: $\|A\|_{\mathcal{X}_k} \leq \|B\|_{\mathcal{X}_k} + \|C\|_{\mathcal{X}_k}$. Thus it suffices to bound each of these terms by 1.

$$\|B\|_{\mathcal{X}_k} \leq \sup_{M: \sum_{i=1}^{d} |M_{i,i}| \leq 1} \langle B, M \rangle = \|B\|_{\infty} \leq \eta_1,$$

where we use that $B$ is a diagonal matrix with entry at most $\eta_1$.

$$\|C\|_{\mathcal{X}_k} \leq \sup_{M: \|M\|_1 \leq k} \langle C, M \rangle = \sup_{M: \|M\|_1 \leq k} \|C\|_{\infty} \|M\|_1 \leq k\eta_2.$$

$\qquad\square$

## A.4  Truncation

We show how truncation can increase the spectral norm of covariance from 1 to $\omega(1)$.

Consider the distribution which, with probability $1/(2k)$, returns a vector where each coordinate is independent $-\sqrt{k}$ with probability $2/3$ and $2\sqrt{k}$ with probability $1/3$. Otherwise, with probability $1 - 1/(2\sqrt{k})$, the distribution returns the origin. The mean of the distribution is the origin, and the covariance is $I$.

Now consider the truncation $h_{0,\sqrt{k}}$, which truncates at distance $\sqrt{k}$ from the origin. Let $Y$ be the resulting random variable. The mean of $Y$, $\mu'$, is thus equal to $(1/2k)(-\sqrt{k}/3, \ldots, -\sqrt{k}/3) = -1/(6\sqrt{k})\mathbf{v}$, where $\mathbf{v}$ is the all ones vector. The norm of $\mu'$ is $\Theta(\sqrt{d/k})$. Since the distribution returns the origin with constant probability (asymptotically tending to 1), the variance of $Y$ along the direction of $\mu'$, which is $\mathbf{v}/\sqrt{d}$, is at least $\Omega(d/k) = \omega(1)$.

# B Concentration and Truncation

**Example 1.** *For any number of moments $t \geq 2$, there is a distribution $X$ satisfying the following conditions: (i) The mean of $X$ is 0, and for every unit vector $v$, the $t^{th}$ moment in direction $v$ is upper bounded by 1, that is, $\mathbb{E}[|\langle v, x \rangle|^t] \leq 1$ for $t \geq 2$, (ii) If $S$ is an arbitrary set of $n \leq o(d^{2/t})$ points from the support of $X$, then the set $S$ cannot be $(\epsilon, O(\sqrt{\epsilon}), k)$-stable, for any $\epsilon > 0$, with respect to the mean of the distribution. As a corollary, no subset of $S$ can be stable either.*

*Proof.* For $j \in [d]$, let $e_j$ be the vector that is 1 on the $j$-th coordinate and 0 otherwise. For a fixed $r$, consider the distribution $P$, supported uniformly on the $2d$ points $S = \{\pm re_1, \pm re_2, \ldots, \pm re_d\}$.

It follows that $P$ is a zero mean distribution. The covariance of the distribution $P$ is $\sum_j (1/d) r^2 e_i e_i^\top = (r^2/d)I$.

Furthermore, for any unit vector $v$ and $t \geq 2$, we have that the $t$-th moment in the direction $v$ is bounded as follows:

$$\mathbb{E}[|v \cdot X|^t] = \sum_{j=1}^{d} \frac{1}{d}|v_j|^t r^t = \frac{r^t}{d}\|v\|_t^t \leq \frac{r^t}{d}\|v\|_2^t \leq \frac{r^t}{d},$$

where we use that $t \geq 2$ and $\|v\|_t \leq \|v\|_2$ for any vector $v$. Thus, we choose $r = d^{1/t}$ for the distribution.

Now we show the second claim, that *any* set of at most $\Omega(d^{2/t})$ samples from this distribution cannot be stable.

Let $S$ be any (multi-)set of $n$ points from the support of $X$. Let $x_1 \in S$. Since $x_1$ is 1-sparse and has $\ell_2$ norm $r$, we have that $x_1 x_1^\top /r^2$ belongs to $\mathcal{X}_k$. Thus we have the following:

$$\left\|\frac{1}{n}\sum_{i \in S'} x_i x_i^\top\right\|_{\mathcal{X}_k} \geq \left\langle \frac{1}{n}\sum_{i \in S'} x_i x_i^\top, \frac{1}{r^2}x_1 x_1^\top \right\rangle \geq \frac{\|x_1\|^4}{r^2 n} = \frac{r^2}{n}$$

Thus, for $r^2/n$ to be upper bounded by a constant, $n$ has to be $\Omega(d^{2/t})$. $\qquad\square$

**Lemma 3.1** (Truncation in $\ell_\infty$). *Let $P$ be a distribution over $\mathbb{R}^d$ with mean $\mu_P$ and covariance $\Sigma_P$, with $\|\Sigma\|_{\mathcal{X}_k} \leq \sigma^2$ for some $\sigma^2 > 0$. Let $X \sim P$ and assume that for all $j \in [d]$, $\mathbb{E}[(X - \mu_P)_j^4] \leq \sigma^4 \nu^4$ for some $\nu \geq 1$. Let $b \in \mathbb{R}^d$ be such that $\|b - \mu\|_\infty \leq a/2$ and $a := 2\sigma\sqrt{k/\epsilon}$ for some $\epsilon \in (0, 1)$. Define $Q$ to be the distribution of $Y := h_{a,b}(X)$. Let the mean and covariance of $Q$ be $\mu_Q$ and $\Sigma_Q$ respectively. Then the following hold:*

*(1) $\|\mu_P - \mu_Q\|_\infty \leq \sigma\sqrt{\epsilon/k}$*

*(2) $\|\mu_P - \mu_Q\|_{2,k} \leq \sigma\sqrt{\epsilon}$*

*(3) $\|\Sigma_P - \Sigma_Q\|_{\mathcal{X}_k} \leq 3\sigma^2 \epsilon \nu^4$*

*(4) For all $i \in [d]$, $\mathbb{E}[(Y - \mu_Q)_i^4] \leq 8\nu^4 \sigma^4$*

*(5) $\|Y - \mu_Q\|_\infty \leq 2a = 4\sigma\sqrt{k/\epsilon}$ almost surely.*

*Proof.* Let $Y := h_{a,b}(X)$ and denote $\mu := \mu_P$. Fix a $i \in [d]$. Since $|\mu_i - b_i| \leq a/2$ and we threshold at the radius $a$, we have the following:

$$|Y_i - \mu_i| \leq |X_i - \mu_i|, \text{ and } |X_i - Y_i| \leq |X_i - \mu_i|. \tag{5}$$

Let $\mathcal{E}_i$ be the event that $Y_i \neq X_i$. We get the following by Markov's inequality and moment bounds:

$$\mathcal{P}(\mathcal{E}_i) = \mathbb{P}(|X_i - b_i| > a) \leq \mathbb{P}(|X_i - \mu_i| \geq a/2) \leq \min\left(4\frac{\sigma^2}{a^2}, 16\frac{\sigma^4 \nu^4}{a^4}\right) = \min\left(\frac{\epsilon}{k}, \frac{\epsilon^2 \nu^4}{k^2}\right). \tag{6}$$

1. We can verify the following relation using Equation (5):

$$|Y_i - X_i| \leq \mathbb{1}_{\mathcal{E}_i} \cdot (|Y_i - X_i|) \leq \mathbb{1}_{\mathcal{E}_i} \cdot (|X_i - \mu_i|). \tag{7}$$

Applying Cauchy-Scharz on the above inequality gives the desired conclusion:

$$|\mathbb{E}[Y_i] - \mu_i| = |\mathbb{E}[Y_i - X_i]| \leq \mathbb{E}\left[\mathbb{1}_{\mathcal{E}} \cdot (|X_i - \mu_i|)\right] \leq \sqrt{\mathbb{P}(\mathcal{E})}\sqrt{\mathbb{E}\left[|X_i - \mu_i|^2\right]} \leq \sigma\sqrt{\frac{\epsilon}{k}},$$

where we use that variance of $X_i$ is at most $\sigma^2$ and use Equation (6).

2. This follows directly from above.

3. By Lemma A.6, it suffices to show that $\|\Sigma_Q - \Sigma_P\|_\infty \leq 3\sigma^2 \epsilon \nu^4 / k$. Using triangle inequality, we obtain the following:

$$\|\Sigma_P - \Sigma_Q\|_\infty = \left\| \mathbb{E}[(X - \mu_P)(X - \mu_P)^\top] - \mathbb{E}[(Y - \mu_P)(Y - \mu_P)^\top] \right.$$
$$\left. + (\mu_Q - \mu_P)(\mu_Q - \mu_P)^\top \right\|_\infty$$
$$\leq \left\| \mathbb{E}[(X - \mu_P)(X - \mu_P)^\top] - \mathbb{E}[(Y - \mu_P)(Y - \mu_P)^\top] \right\|_\infty$$
$$+ \left\| (\mu_Q - \mu_P)(\mu_Q - \mu_P)^\top \right\|_\infty.$$

By the first part above, we have that $\|(\mu_Q - \mu_P)(\mu_Q - \mu_P)^\top\|_\infty \leq \sigma^2 \epsilon / k \leq \sigma^2 \nu^4 \epsilon / k$, where we use that $\nu \geq 1$. We will thus focus on the first term. Without loss of generality, we will assume that $\mu_P = 0$ for the remainder of this proof. Thus for any $i, j \in [d]$, we thus need to upper bound $\mathbb{E}[|X_i X_j - Y_i Y_j|]$.

$$\mathbb{E}[|X_i X_j - Y_i Y_j|] \leq \mathbb{E}[|X_i||X_j - Y_j|] + \mathbb{E}[|Y_j||X_i - Y_i|]$$
$$\leq \mathbb{E}[|X_i||X_j| \cdot \mathbb{1}_{\mathcal{E}_j}] + \mathbb{E}[|X_i||X_j| \cdot \mathbb{1}_{\mathcal{E}_i}] \qquad \text{(Using Equation (7))}$$
$$\leq \sqrt{\mathbb{E}[|X_i X_j|^2]} \left( \sqrt{\mathbb{P}(\mathcal{E}_i)} + \sqrt{\mathbb{P}(\mathcal{E}_j)} \right)$$
$$\leq (\mathbb{E}[X_i^4])^{1/4}(\mathbb{E}[X_j^4])^{1/4} \left( \sqrt{\mathbb{P}(\mathcal{E}_i)} + \sqrt{\mathbb{P}(\mathcal{E}_j)} \right)$$
$$= \sigma^2 \nu^2 \left( 2\frac{\epsilon \nu^2}{k} \right)$$
$$= \frac{2\sigma^2 \nu^4 \epsilon}{k}.$$

Combining the above with Lemma A.6, we get that the $\|\Sigma_P - \Sigma_Q\|_{\mathcal{X}_k} \leq 3\sigma^2 \epsilon \nu^4$.

4. Fix an $i \in [d]$. We use the triangle inequality and Equation (7) to get the following:

$$\mathbb{E}[(Y - \mu_Q)_i^4] \leq 4(\mathbb{E}[(Y - \mu_P)_i^4]) + 4\|\mu_P - \mu_Q\|_\infty^4 \leq 4\sigma^4 \nu^4 + 4\sigma^4 \epsilon^2 / k^2 \leq 8\sigma^4 \nu^4,$$

where the last inequality uses that $\nu \geq 1$ and $\epsilon \leq 1$.

5. This follows by definition of the random variable $Y$, the function $h_{a,b}$, and the parameter $a$.

$\square$

**Fact B.1** (VC inequality). *Let $\mathcal{F}$ be a family of boolean functions over $\mathcal{X}$ with VC dimension $r$ and let $S = \{x_1, \ldots, x_n\}$ be a set of $n$ i.i.d. data points from a distribution $P$ over $\mathcal{X}$. If $n \gg c(r + \log(1/\tau))/\gamma^2$, then with probability $1 - \tau$, for all $f \in \mathcal{F}$, we have that*

$$\left| \sum_{i=1}^n \frac{f(x_i)}{n} - \mathbb{E}_P[f(x)] \right| \leq \gamma.$$

**Lemma B.2** (Uniform concentration over $\mathcal{A}_{k,P}$). *Let $S$ be a set of $n$ i.i.d. data points from a distribution $P$, and let $\mathcal{A}_{k,P}$ be as defined in Equation (9). There exists a constant $c > 0$ such that if $n \geq c(k^2 \log d + \log(1/\tau))/(q^2)$, then Equation (10) holds with probability at least $1 - \tau$ over the set $S$ of $n$ i.i.d. points from distribution $P$.*

*Proof.* Let $Q$ be the distribution of $y := xx^\top$. Let $\mathcal{F} := \{\mathbb{1}_{y \cdot A > s_1} : A \in \mathcal{A}_k\}$. Suppose for now that VC dimension of $\mathcal{F}$ is less than $Ck^2 \log(d)$. Then the standard VC inequality (Fact B.1) implies that if $n \geq c(k^2 \log d + \log(1/\tau))/(q^2)$, then Equation (10) holds because under $y \sim Q$, $\mathbb{P}(y \cdot A > s_1) \leq q$ for all $A \in \mathcal{A}_{k,P}$. Thus it remains to show an upper bound on the VC dimension of $\mathcal{F}$. Since $\mathcal{F}$ corresponds to a family of linear functions that are $k^2$-sparse in $d^2$ dimensional space, [AV19, Theorem 6] implies that the VC dimesion is at most $4k^2 \log(3d)$. This completes the proof. $\qquad\square$

## C   Stability with High Probability

**Theorem C.1.** *Let $S$ be a set of $n$ i.i.d. data points from a distribution $P$ over $\mathbb{R}^d$. Let the mean of $P$ be $\mu$, and covariance $\Sigma$ such that $\|\Sigma\|_{\mathcal{X}_k} \leq \sigma^2$, and for all $j \in [d]$, $\mathbb{E}[(X_j - \mu)^4] \leq \nu^4$. Suppose $P$ is supported over the set $\{x : \|x - \mu\|_\infty \leq \sigma \times r \times \sqrt{k}\}$. If $n = \Omega(k^2 \log d + \log(1/\tau))$, then, with probability $1 - \tau$, there exists a set $S' \subset S$ such that:*

1. *$|S'| \geq 0.98n$*

2. *$S'$ is $(0.01, \delta, k)$-stable with respect to $\mu$ and $\sigma$ where $\delta = O(\max(1, r^2, \nu^2/\sigma^2))$.*

*Proof.* In the following proof, we will use notations $q, s_1, s_2, s_3, V_Z$ and $B$, all of which are either constants or functions of $\sigma$, $r$ and $\nu$ in the theorem statement. The functions are explicitly chosen in Appendix C.1.

We will assume $\mu = 0$ without loss of generality. Instead of directly showing the existence of subset $S' \subseteq S$ (with high probability over the samples $S$) that is stable, Proposition A.1 in Appendix A lets us show the following simpler condition: let $\Delta_{n,\epsilon}$ be the set of weights/distributions $w$ such that $w_i \leq 1/(1 - \epsilon)$, then there exists a weighting $w \in \Delta_{n,0.01}$ such that $\|\Sigma_w\|_{\mathcal{X}_k} \leq B$ for the function $B$ chosen in Appendix C.1, which satisfies $B = O(\sigma^2 \max(1, r^2, \nu^2/\sigma^2))$. That is, for the following proof, we just need to prove that $\min_{w \in \Delta_{n,0.01}} \|\Sigma_w\|_{\mathcal{X}_k} \leq B$.

We proceed as follows:

$$\min_{w \in \Delta_{n,0.01}} \|\Sigma_w\|_{\mathcal{X}_k} = \min_{w \in \Delta_{n,0.01}} \max_{M \in \mathcal{X}_k} \langle M, \Sigma_w \rangle = \max_{M \in \mathcal{X}_k} \min_{w_M \in \Delta_{n,0.01}} \langle M, \Sigma_w \rangle$$

where the last equality is a straightforward application of the minimax theorem for a minimax optimization problem with independent convex domains and a bilinear objective. It thus suffices to show the following: with probability $1 - \tau$,

$$\forall M \in \mathcal{X}_k : \ |\{x \in S : x_i^\top M x_i > B\}| \leq 0.01|S| \tag{8}$$

from which we can construct the weighting $w_M$ as uniform distribution over the elements outside the above set.

Define the following sets of sparse matrices:

$$\mathcal{A}_k := \left\{ A \in \mathbb{R}^{d \times d} : \|A\|_0 \leq k^2, \|A\|_F \leq 1 \right\},$$
$$\mathcal{A}_{k,P} := \left\{ A \in \mathcal{A}_k : \mathbb{P}\left\{ x^\top A x \geq s_1 \right\} \leq q \right\}. \tag{9}$$

where $q$ and $s_1$ are chosen in Appendix C.1. If $n \gtrsim (k^2 \log d + \log(1/\tau))/(q^2)$, then a standard covering/VC-dimension bound (see Lemma B.2 for details) implies that the following event holds with probability $1 - \tau$:

$$\forall A \in \mathcal{A}_{k,P} : \ |\{x \in S : x_i^\top A x_i > s_1\}| \leq 2 \times q \cdot |S|. \tag{10}$$

Our choice of $q$ is a constant (cf. Appendix C.1) and thus the required sample complexity for Equation (10) to hold is $\Omega(k^2 \log d + \log(1/\tau))$. We will now show that the event in Equation (10) implies that the event in Equation (8) holds.

Suppose, for the sake of contradiction, that the event in Equation (8) does not hold. Then there exists a $M \in \mathcal{X}_k$ such that $|\{x \in S : x_i^\top M x_i > B\}| > 0.01|S|$. We will show the existence of a matrix $Q$ violating event Equation (10), via the probabilistic method, to reach the desired contradiction.

Fixing an $M$ that violates Equation (8), consider the random matrix $Q$ where each entry $Q_{i,j}$ is sampled independently from the following distribution, defined using the constant $s_2$ chosen in Appendix C.1:

$$Q_{i,j} := \begin{cases} M_{i,j}, & \text{with prob. 1 if } |M_{i,j}| \geq s_2/k, \\ \frac{s_2}{k}\operatorname{sign}(M_{i,j}), & \text{with prob. } |kM_{i,j}|/s_2 \text{ if } |M_{i,j}| \leq s_2/k, . \\ 0, & \text{with remaining prob. if } |M_{i,j}| \leq s_2/k \end{cases} \tag{11}$$

Defining $p_{i,j}$ to be $\min(1, k|M_{i,j}|/s_2)$, then $Q_{i,j}$ is equivalently $M_{i,j}/p_{i,j}$ with probability $p_{i,j}$ and 0 otherwise.

We will show that the following events hold simultaneously with non-zero probability, leading to a contradiction to event Equation (10):

1. $Q \in s_3\mathcal{A}_{k,P}$

2. $|\{x \in S : x_i^\top Q x_i > s_3 \times s_1\}| > 2 \times q \cdot |S|$

where $s_3$ is also a constant, larger than 2, and explicitly chosen in Appendix C.1. Using different techniques, we will show that the first condition holds with probability at least $1 - 2 \times 10^{-6}$ and the second condition holds with probability at least $4 \times 10^{-6}$, thus implying that the events hold simultaneously with non-zero probability.

**Condition 1** We begin with the following lemma showing that $Q$ lies in $s_3\mathcal{A}_k$ with high probability.

**Lemma C.2** ($Q$ lies in $s_3\mathcal{A}_k$ with high probability). *Let $Q$ be generated as described in Equation (11), for an $M \in \mathcal{X}_k$. Then with probability except $(1/s_2) + (s_2/s_3^2)$, we have that that $Q \in s_3 A_k$.*

*Proof.* The expected sparsity of $Q$ is at most $\sum_{i,j} \frac{k}{s_2}|M_{i,j}| \leq \frac{k^2}{s_2}$ since $|M|_1 \leq k$. Thus, by Markov's inequality, except with $1/s_2$ probability, $Q$ and hence $Q/s_3$ is $k^2$-sparse. We also have to show that with probability at least $1 - 10^8$, $\|Q\|_F \leq s_3$.

$$\mathbb{E}\|Q\|_F^2 \leq \sum_{i,j} \left(\frac{s_2}{k}\right)^2 \left(\frac{k|M_{i,j}|}{s_2}\right) = \frac{s_2|M_{i,j}|}{k} = s_2. \tag{12}$$

Again, by Markov's inequality, we get that with probability except $s_2/s_3^2$, the Frobenius norm of $Q$ is at most $s_3$. The lemma statement follows from the union bound. $\qquad \square$

Our choice of constants in Appendix C.1 would ensure that the failure probability in Lemma C.2 is at most $10^{-6}$.

$$\frac{1}{s_2} + \frac{s_2}{s_3^2} \leq 10^{-6}. \tag{13}$$

It remains to show that $Q$ belongs to $s_3\mathcal{A}_{k,p}$ with high (constant) probability, i.e., with probability $10^{-6}$ over sampling of $Q$, we have that $\mathbb{P}_{x \sim P}(x^\top Q x > s_3 \times s_1 | Q) \leq q$. Let $R := x^\top Q x$, where both $x$ and $Q$ are sampled independently from $P$ and Equation (11) respectively.

To show this, we use the following the lemma for a sufficient condition involving sampling both $x$ and $Q$.

**Lemma C.3.** *Consider a probability space over the randomness of independent variables $X$ and $Y$. Suppose the event $E$ (over pairs $(X, Y)$) happens with probability at least $1 - \alpha\beta$ for some $\alpha, \beta \in [0, 1]$. Then, it must be the case that, with probability at least $1 - \alpha$ over the sampling of $X$, the conditional probability of $E$ given $X$ is at least $1 - \beta$.*

*Proof.* For the sake of contradiction, suppose the lemma conclusion is false. Then

$$\mathbb{P}_{X,Y}(E) = \int_Y \mathbb{P}(E|X)\, d\,\mathbb{P}(X) < (1 - \alpha) + \alpha(1 - \beta) = 1 - \alpha\beta,$$

which contradicts the premise. $\qquad \square$

It thus suffices to show that with probability $1 - 10^{-6} \times q$ over both $x$ and $Q$, $R \leq s_3 \times s_1$.

**Lemma C.4.** *Let $R = x^\top Q x$, where $Q$ is independently drawn from the distribution in Equation (11) and $x$ is drawn independently from $P$. Under the assumptions of Theorem C.1,*

$$\mathbb{P}\{R > s_3 \times s_1\} \leq \frac{\sigma^2}{s_1} + \frac{4}{s_3} + \frac{s_2 \times \nu^4}{s_3 \times s_1^2}, \tag{14}$$

*Proof.* We consider three exhaustive events, over $x$ and $Q$, of $\mathcal{E} := \{R > s_3 \times s_1\}$, and bound the probability of each of them :

1. $\mathcal{E}_1 := \{(x, Q) : \mathbb{E}[R|x] > s_1\}$. Since $\mathbb{E}[R|x] = x^\top M x$, the event corresponds to $\{x : x^\top M x > s_1\}$. We have that $\mathbb{E}[x^\top M x] = \langle \Sigma, M \rangle \leq \|\Sigma\|_{\mathcal{X}_k} = \sigma^2$. By Markov's inequality, $\mathbb{P}(\mathcal{E} \cap \mathcal{E}_1) \leq \mathbb{P}(\mathcal{E}_1) \leq \sigma^2/(s_1)$.

2. $\mathcal{E}_2 := \{(x, Q) : x \in \mathcal{F}\}$, where $\mathcal{F}$ is the following event over $x$: $\mathcal{F} = \{x : \mathbb{E}[R|x] \leq s_1, \mathrm{Var}(R|x) \leq s_3 \times s_1^2\}$. Observe that conditioned on $x \in \mathcal{F}$, we have that $R|x$ is a random variable with mean at most $s_1$ and variance at most $s_3 \times s_1^2$. Thus for each such $x \in \mathcal{F}$, the conditional probability that $R > s_3 \times s_1$ is at most $s_3 s_1^2/((s_3 - 1)^2 s_1^2)$ by Chebyshev's inequality. We thus get that $\mathbb{P}(\mathcal{E}_2 \cap \mathcal{E}) \leq \mathbb{P}(\mathcal{E}|\mathcal{E}_2) = \mathbb{P}(\mathcal{E}|x \in \mathcal{F}) \leq 4/(s_3)$, where we use that $s_3 \geq 2$.

3. $\mathcal{E}_3 := \{(x, Q) : \mathrm{Var}(R|x) \geq s_3 \times s_1^2\}$. We will upper bound $\mathbb{P}(\mathcal{E}_3)$. We first calculate the $\mathrm{Var}(R|x)$ using the independence of entries of $Q$ as follows:

$$\mathrm{Var}(R|x) = \sum_{i,j} x_i^2 x_j^2 \mathrm{Var}(Q_{i,j}) = \sum_{i,j:|M_{i,j}| \leq s_2/k} x_i^2 x_j^2 |M_{i,j}| \left( \frac{s_2}{k} - |M_i, j| \right).$$

To show that $\mathrm{Var}(R|x)$ is small with high probability, we will upper bound $\mathbb{E}[\mathrm{Var}(R|x)]$.

$$\begin{aligned}
\mathbb{E}[\mathrm{Var}(R|x)] &= \sum_{i,j:|M_{i,j}| \leq s_2/k} |M_{i,j}| \left( \frac{s_2}{k} - |M_i, j| \right) \mathbb{E}[x_i^2 x_j^2] \\
&\leq \sum_{i,j} \frac{s_2}{k} |M_{i,j}| \mathbb{E}[x_i^2 x_j^2] \\
&\leq \frac{s_2 \times \|M\|_1 \times \nu^4}{k} \qquad \text{(using } \mathbb{E}[x_i^2 x_j^2] \leq \sqrt{\mathbb{E}[x_i^4] \mathbb{E}[x_j^4]} = \nu^4) \\
&\leq s_2 \times \nu^4. \qquad\qquad\qquad\qquad\qquad \text{(using } \|M\|_1 \leq k)
\end{aligned}$$

Thus Markov's inequality implies that $\mathbb{P}(\mathcal{E} \cap \mathcal{E}_3) \leq \mathbb{P}(\mathcal{E}_3) \leq (s_2 \times \nu^4)/(s_3 \times s_1^2)$.

Taking the union bound, we get the desired result. $\qquad\square$

As reasoned above, we want the failure probability in Equation (14) to be less than $10^{-6} \times q$. That is,

$$\frac{\sigma^2}{s_1} + \frac{4}{s_3} + \frac{s_2 \times \nu^4}{s_3 \times s_1^2} \leq 10^{-6} \times q. \tag{15}$$

In Appendix C.1, we choose $s_1, s_2, s_3$ and $q$ such that the bound holds. This, by the reasoning after Lemma C.3, guarantees that $Q$ satisfies the extra condition for $s_3 \mathcal{A}_{k,p}$ (on top of being in $s_3 \mathcal{A}_k$) with probability at least $1 - 10^{-6}$.

Taking a union bound, with failure probabilities $10^{-6}$ (for $Q$ being in $s_3 \in \mathcal{A}_k$, Lemma C.2) and $10^{-6}$ for satisfying the additional criterion for being in $s_3 \mathcal{A}_{k,p}$, we conclude that Condition 1 happens with probability $1 - 2 \cdot 10^{-6}$.

**Condition 2** The second condition makes crucial use of the Berry-Esseen theorem, which we restate as follows.

**Fact C.5** (Berry-Esseen Theorem for sums of independent variables). *Consider a random variable $\xi = \sum_i \xi_i$, where the variables $\xi_i$ are independent (but not necessarily identical) and each of them has finite third moment. Denote $\mu_i$ as $\mathbb{E}[\xi_i]$, $\sigma_i^2$ as $\mathrm{Var}(\xi_i)$ and $\rho_i$ as the third central absolute moment, namely $\rho_i = \mathbb{E}[|\xi_i - \mu_i|^3]$. Then,*

$$d_{\mathrm{K}}(\xi, \mathcal{N}(\sum_i \mu_i, \sigma_i^2)) \leq 0.57 \frac{\sum_i \rho_i}{(\sum_i \sigma_i^2)^{1.5}} = 0.57 \frac{\sum_i \rho_i}{(\mathrm{Var}(\xi))^{1.5}}$$

*where $d_{\mathrm{K}}$ is the Kolmogorov distance between two distributions (namely, the $\ell_\infty$ distance between the cumulative density functions).*

Define the random variable $Z$ to be

$$Z = \sum_i \mathbb{1}_{(x_i x_i^\top) \bullet Q > s_3 \times s_1}. \tag{16}$$

The second condition is equivalent to saying that $Z > 2 \times q \times |S|$, which we show to happen with probability at least $4 \times 10^{-6}$.

The strategy is to lower bound $\mathbb{E}[Z]$, and then use Paley-Zygmund to show that $Z$ is large with constant probability. To lower bound the expectation, for any $i$ such that $(x_i x_i^\top) \bullet M > B$, we want to lower bound $\mathbb{P}_Q((x_i x_i^\top) \bullet Q > s_3 \times s_1)$, using either Chebyshev's inequality or the Berry-Esseen theorem. First, note that for these $i$, $\mathbb{E}[(x_i x_i^\top) \bullet Q] = (x_i x_i^\top) \bullet M > B$ by our assumption. If $\mathrm{Var}[(x_i x_i^\top) \bullet Q] \leq V_Z$, where $V_Z$ is a fixed function of $r$ and $\sigma$ chosen in Appendix C.1, then by Chebyshev's inequality, we have

$$\mathbb{P}\left((x_i x_i^\top) \bullet Q \geq s_3 \times s_1\right) \geq \mathbb{P}\left((x_i x_i^\top) \bullet Q \geq B - 10 \times \sqrt{V_Z}\right) \geq 0.99. \tag{17}$$

where the first inequality is true by our choice of $s_1, s_3, V_Z$ and $B$ in Appendix C.1. Otherwise, we have the case where $\mathrm{Var}[(x_i x_i^\top) \bullet Q] > V_Z$. In this case, we treat $(x_i x_i^\top) \bullet Q$ as a sum of independent variables

$$(x_i x_i^\top) \bullet Q = \sum_{s,t} (x_i)_s (x_i)_t Q_{s,t}$$

and use the Berry-Esseen theorem, which requires bounding the sum of the third central absolute moment of the summands. Let $\rho_{s,t}$ be the third central absolute moment of $(x_i)_s (x_i)_t Q_{s,t}$. For any $(s,t)$ such that $0 < |M_{s,t}| \leq s_2/k$, we can calculate its third moment as follows:

$$\begin{aligned}
\rho_{s,t} &= \mathbb{E}[|(x_i)_s (x_i)_t Q_{s,t} - \mathbb{E}[(x_i)_s (x_i)_t Q_{s,t}]|^3] \\
&= |(x_i)_s|^3 |(x_i)_t|^3 \frac{|M_{s,t}|^3}{p_{s,t}^3} \mathbb{E}[|\mathrm{Ber}(p_{s,t}) - p_{s,t}|^3] \\
&= |(x_i)_s|^3 |(x_i)_t|^3 \frac{|M_{s,t}|^3}{p_{s,t}^3} p_{s,t}(1 - p_{s,t})(1 - 2p_{s,t} + 2p_{s,t}^2) \\
&\leq |(x_i)_s|^3 |(x_i)_t|^3 \frac{|M_{s,t}|^3}{p_{s,t}^3} p_{s,t}(1 - p_{s,t}) && \text{for all } p_{s,t} \in [0,1] \\
&\leq (\sigma^2 \times r^2 \times s_2)(x_i)_s^2 (x_i)_t^2 \frac{M_{s,t}^2}{p_{s,t}^2} p_{s,t}(1 - p_{s,t}) \\
&&&\hspace{-3cm}\text{since } |x_i|_\infty \leq \sigma \times r \times \sqrt{k} \text{ and } |M_{s,t}|/p_{s,t} = s_2/k
\end{aligned}$$

The same inequality holds trivially for $(s,t)$ where $|M_{s,t}| \geq s_2/k$ or $M_{s,t} = 0$ since $\rho_{s,t} = 0$ in both of these edge cases. Thus, the sum of the third central absolute moment of the summands we need for Berry-Esseen is

$$\begin{aligned}
\sum_{s,t} \rho_{s,t} &\leq (\sigma^2 \times r^2 \times s_2) \sum_{s,t} (x_i)_s^2 (x_i)_t^2 \frac{M_{s,t}^2}{p_{s,t}^2} p_{s,t}(1 - p_{s,t}) \\
&= (\sigma^2 \times r^2 \times s_2) \mathrm{Var}\left((x_i x_i^\top) \bullet Q\right)
\end{aligned}$$

where the last equality is a simple calculation to calculate the term-by-term variance for $(x_i x_i^\top) \bullet Q$. Thus, Fact C.5 implies that the Kolmogorov distance between the distribution of $(x_i x_i^\top) \bullet Q$ and the Gaussian with the same mean and variance is at most

$$0.57 \frac{\sum_{s,t} \rho_{s,t}}{(\text{Var}((x_i x_i^\top) \bullet Q))^{1.5}} \leq 0.57(\sigma^2 \times r^2 \times s_2) \frac{\text{Var}\left((x_i x_i^\top) \bullet Q\right)}{\text{Var}^{1.5}\left((x_i x_i^\top) \bullet Q\right)} \leq \frac{0.57(\sigma^2 \times r^2 \times s_2)}{\sqrt{V_Z}}$$

where the inequality comes from the assumption that the variance is at least $V_Z$. Therefore, $(x_i x_i^\top) \bullet Q$ has at least probability $0.5 - \frac{0.57(\sigma^2 \times r^2 \times s_2)}{\sqrt{V_Z}}$ of exceeding its expectation. By our choice of quantities in Appendix C.1, this probability is at least $0.4$. Furthermore, $\mathbb{E}((x_i x_i^\top) \bullet Q) = x_i x_i^\top \bullet M$ is bigger than $B$ and in turn bigger than $s_3 \times s_1$ (by our choice for these quantities). Thus, with probability at least $0.4$, $(x_i x_i^\top) \bullet Q$ exceeds $s_3 \times s_1$.

Combined with the guarantee that $\mathbb{P}_Q((x_i x_i^\top) \bullet Q > s_3 \times s_1) > 0.99$ in the case where $\text{Var}((x_i x_i^\top) \bullet Q) \leq V_Z$ (cf. Equation (17)), we have shown that in all cases, $\mathbb{P}_Q((x_i x_i^\top) \bullet Q > s_3 \times s_1) > 0.4$ whenever $x_i x_i^\top \bullet M > B$.

Thus, we have shown that $\mathbb{E}[Z] = \sum_i \mathbb{P}_Q((x_i x_i^\top) \bullet Q > s_3 \times s_1) > 0.4 \times 0.01 n = 0.004 n$, since at least $0.01$ fraction of points satisfy $x_i x_i^\top \bullet M > B$ and thus also satisfy $\mathbb{P}_Q((x_i x_i^\top) \bullet Q > s_3 \times s_1) > 0.4$. Note also that $Z \in [0, n]$ always, meaning that $\mathbb{E}[Z^2] \leq n^2$. Since $0.004 \geq 4 \times q$ by our choice of $q$, it then follows from the Paley-Zygmund inequality that

$$\mathbb{P}(Z > 2 \times q \times |S|) \geq 0.25 \frac{(\mathbb{E}[Z])^2}{n^2} > \frac{0.25 \times 0.004^2 n^2}{n^2} = 4 \cdot 10^{-6}$$

showing the second claim above, and completing the proof of this lemma.

$\square$

## C.1   Choice of Numerical Constants

This section shows how to pick the numerical constants $q, s_1, s_2, s_3, V_Z$ and $B$. In the proof of Theorem C.1, these constants need to satisfy the following constraints:

1. $s_3 \geq 2$
2. $q$ is at least a small constant since the sample complexity is inversely proportional to $1/q^2$.
3. See (13)

$$\frac{1}{s_2} + \frac{s_2}{s_3^2} \leq 10^{-6}$$

4. See (15)

$$\frac{\sigma^2}{s_1} + \frac{4}{s_3} + \frac{s_2 \times \nu^4}{s_3 \times s_1^2} \leq 10^{-6} \times q$$

5. See (17) $B \geq s_3 \times s_1 + 10\sqrt{V_Z}$
6. See (18) $\frac{0.57(\sigma^2 \times r^2 \times s_2)}{\sqrt{V_Z}} \leq 0.1$
7. Paley-Zygmund $0.004 \geq 4 \times q$

Therefore, we pick the constants as follows:

1. $\nu, \sigma$ and $r$ are numbers we get from the $\ell_\infty$ truncation, nothing to choose here.
2. $q = 0.001$
3. $s_2 = 10^7$
4. $s_3 = 10^{10}$
5. Solve for $s_1$ in terms of above in Constraint 4. Suffices to take $s_1 = \max(\sigma^2, \nu^2) \times 10^{10}$
6. Solve for $\sqrt{V_Z}$ using Constraint 6. Suffices to take $V_Z = 10^{16}\sigma^4 r^4$.
7. Solve for $B$ using Constraint 5. Suffices to take $B = \max(\sigma^2, \sigma^2 r^2, \nu^2) \times 10^{20}$.

# D   Smoothness of Stability

The goal of this appendix is to prove Theorem D.11, the stability result we use in the proof of Theorem 1.3.

In Section 5, we sketched the proof of Theorem 5.1, which we formalized as Theorem C.1 in Appendix C. The key difference between Theorems C.1 and D.11 is that the former is a stability result concerning uncontaminated samples truncated according to some *fixed* vector close to the true mean. On the other hand, Theorem D.11 concerns samples truncated according to the coordinate-wise median-of-means estimate, which itself depends on the samples and is not fixed. Thus, much of this appendix is dedicated to showing the "Lipschitzness" of the stability of samples, as we truncate using different preliminary mean estimates.

We start with showing Lemma D.1, which states that with high probability, there exists a large subset of samples where in each dimension, at most a negligible fraction of the points have large magnitude. Then, Lemma D.9 shows that we can take the intersection between this large subset and the large subset of samples that are stable. Finally, Lemma D.10 and Theorem D.11 show that this subset is stable as long as the truncation is centered at any point close to the true mean, thus yielding the final stability result we desire.

For a vector $X_i \in \mathbb{R}^d$, we will use $X_{i,j}$ to refer to the $j$-th coordinate of $X_i$.

**Lemma D.1.** *Let $P$ be a distribution over $\mathbb{R}^d$. For $X \sim P$, suppose for all $j \in [d]$, $\mathbb{E}[X_j^4] \leq \nu^4$. Then let $S$ be a set of $n$ i.i.d. points from $P$. Then there exists positive constants $c_1$ such that the following holds with probability $1 - \tau$ if $n \geq c_1(k^{1.5} + \log(1/\tau))$: there is a set $S' \subset S$ such that the following hold simultaneously:*

1. *$|S'| \geq 0.99|S|$*

2. *For each $j$ in $[d]$, the number of points in $S'$ with $j$-th coordinate larger in magnitude than $2\nu\sqrt{k}$ is at most $n/k^{1.5}$. Equivalently,*

$$\forall j \in [d] : \quad \sum_{i \in S} \mathbb{1}_{|X_{i,j}| \geq 2\nu\sqrt{k}} \leq \frac{n}{k^{1.5}} \tag{18}$$

Before providing the proof of Lemma D.1, we highlight why the result is not obvious. The first approach that one may try is to show that the original set $S$ directly satisfies the claim, i.e., (with high probability) in each coordinate, the fraction of points with large magnitude in that coordinate is at most $k^{-1.5}$. At the population level, this is indeed true by the fourth moment assumption, i.e., for any fixed $i \in [n]$ and $j \in [d]$, the probability that $|X_{i,j}|$ is large is at most $O(1/k^2)$. However, for this to hold with probability $1 - \tau$, one requires roughly $k^{1.5}\log(1/\tau)$ samples even in 1 dimension[3], which would give a multiplicative dependence on $\log(1/\tau)$ instead of additive dependence.

The second approach that one may try would be the following: define $S'$ to be the set of all "good" samples, where we say a sample is "good" if all of its coordinates are smaller than $c\nu\sqrt{k}$. For any fixed coordinate $j \in [d]$, the probability that the $j$-th coordinate may be larger than $c\nu\sqrt{k}$ may be as large as $1/k^2$. Thus the probability that a particular sample is bad may be arbitrarily close to $1$ — for example, when coordinates are independent — and the resulting set $S'$ will be too small with high probability.

*Proof.* We will assume that $k \geq C$ for a large enough constant. If $k$ is smaller than the constant, then the result follows by applying Bernstein inequality and taking $S' = S$.

---

[3]The upper bound follows from a Chernoff bound, and the lower bound follows from the fact that Chernoff bounds are essentially tight for Bernoulli coins.

Let $S = \{Y_1, \ldots, Y_n\}$. For $i \in [n]$ and $j \in [d]$, we use $Z_{i,j}$ to denote $\mathbb{I}_{|Y_{i,j}| \geq c_2 \nu \sqrt{k}}$. For simplicity, we set $\alpha = k^{-1.5}/3$. Our goal is to show that the following integer program is feasible:

$$
\begin{aligned}
\text{variables} \quad & p_1, \ldots, p_n \\
\text{subject to} \quad & \forall j \in [d] : \sum_{i=1}^{n} p_i Z_{i,j} \leq 3\alpha n \\
& \sum_{i=1}^{n} p_i \geq 0.99n \\
& \forall i \in [n] : p_i \in \{0, 1\}.
\end{aligned}
\tag{F1}
$$

As argued above in the prose after the statement, one needs to argue about all the samples, and their coordinates, simultaneously to prove the statement. Since directly handling the feasibility program (F1) seems difficult, our argument will go in the following steps: (i) first consider the LP relxation of (F1), (ii) using duality theory, the LP relaxation is feasible iff the dual LP is infeasible, (iii) Simplify the dual LP and show that, with high probability, the resulting program is infeasible.

We begin by considering the LP relaxation.

$$
\begin{aligned}
\text{variables} \quad & p_1, \ldots, p_n \\
\text{subject to} \quad & \forall j \in [d] : \sum_{i=1}^{n} p_i Z_{i,j} \leq \alpha n \\
& \sum_{i=1}^{n} p_i \geq 0.999n \\
& \forall i \in [n] : p_i \in [0, 1].
\end{aligned}
\tag{F2}
$$

We first show that if the following LP relaxation, (F2), is feasible, then (F1) is also feasible.

**Claim D.2** (Feasibility of (F2) implies feasibility of (F1)). *Suppose $n > 10^6$ and $\alpha \geq (4 \log n)/n$. If (F2) is feasible, then (F1) is also feasible.*

*Proof.* Let $p_1, \ldots, p_n$ be the feasible solution to (F2). Consider the following random assignment, for $i \in [n]$, $P_i \sim \text{Ber}(p_i)$ independently. We will show that, with non-zero probability, $P_i$'s satisfy (F1). We will use the following inequality:

**Fact D.3** (Chernoff Inequality). *Let $a_1, \ldots, a_n$ such that $a_i \in \{0, 1\}$. Let $W_1, \ldots, W_n$ be independent Bernoulli random variables and consider the random variable $Z = \sum_{i=1}^{n} W_i$. Then, with probability $1 - \tau$, $Z \leq 2(\mathbb{E}\, Z + \log(1/\tau))$.*

By Fact D.3, we get that the each of the inequalities in (F1) holds with probability $1 - 1/(2n)$ as long as $n\alpha \geq 2 \log(2n)$ and $n > 1000 \log(2n)$. The latter holds when $n \geq 10^6$. $\qquad\square$

Since $n > 10^6$ in our setting (as $k$ is large and choosing $c_1$ to be large enough) and $\alpha = 1/(3k^{1.5})$, we have that $\alpha \geq 4(\log n)/n$ is equivalent to $n \geq 12k^{1.5} \log n$, which is satisfied when $n \geq 100k^{1.5} \log k$. The latter holds when $n \geq ck^{1.5} \log d$ for a large enough constant $c$. Thus in the remainder of this section, we will show that, with high probability, this LP program is indeed feasible. We begin by considering the following dual program:

$$
\begin{aligned}
\text{variables} \quad & w_1, \ldots, w_d, y_1, \ldots, y_n, x \\
\text{subject to} \quad & \sum_{i=1}^{n} y_i + \alpha n \sum_{j=1}^{d} w_j < 0.999nx \\
& \forall i \in [n] : y_i + \sum_{j=1}^{d} Z_{i,j} w_j \geq x \\
& z \geq 0, \quad \forall i \in [n] : y_i \geq 0, \quad \forall j \in [d] : w_j \geq 0.
\end{aligned}
\tag{F3}
$$

Suppose for the sake of contradiction that (F2) is infeasible. By Farkas' Lemma [GKT51], it means that the (dual) program in (F3) is feasible. Formally, we have the following claim:

**Claim D.4** (LP Duality for (F2))**.** *(F3) is infeasible if and only if (F2) is feasible.*

Claim D.4 follows from Farkas' Lemma. We will argue that (F3) is infeasible by showing that the following program, which is feasible whenever (F3) is feasible, is infeasible.

$$
\begin{aligned}
\text{variables} \quad & w_1, \ldots, w_d, A \\
\text{subject to} \quad & \forall i \in A : \sum_{j=1}^d Z_{i,j} w_j \geq \alpha \Big( \sum_{j=1}^d w_j \Big) \\
& \forall j \in [d] : w_j \geq 0, \\
& A \subset [n], |A| \geq 10^{-3} n
\end{aligned}
\tag{F4}
$$

(F4) states that for at least $10^{-3}$ fraction of $i$'s in $n$, the following inequality holds: $\sum_{j=1}^d Z_{i,j} w_j \geq \alpha \|w\|_1$. The following claim relates the two programs above.

**Claim D.5.** *If (F3) is feasible, then (F4) is feasible.*

*Proof.* Let $y_1, \ldots, Y_n, w_1, \ldots, w_d, x$ be any feasible solution to (F3). Then the first constraint in (F3) that the average of $y_i$'s is less than $0.999x - \alpha(\sum_{j=1}^d w_j)$. By Markov's inequality, the fraction of the $y_i$'s such $y_i \geq (x - \alpha(\sum_{j=1}^d w_j))$ is at most $\frac{0.999x - \alpha(\sum_{j=1}^d w_j)}{(x - \alpha(\sum_{j=1}^d w_j))} \leq 0.999$. Thus the fraction of $y_i$'s such that $y_i < (x - \alpha(\sum_{j=1}^d w_j))$ is at least $0.001$.

Let $A \subset [n]$ be the set of such indices. For any $i \in A$, the second constraint in (F3) implies that $\sum_{j=1}^d Z_{i,j} w_j \geq x - y_i \geq \alpha(\sum_{j=1}^d w_j)$. This implies that (F4) is feasible. $\square$

In order to argue that (F4) is infeasible, we first consider a particular $w$. Using calculations provided below, it can be seen that the probability that a particular $w$ satisfies (F4) is exponentially small in $n$. However, a direct approach at covering $w$ seems difficult since $w$ is a dense vector in $\mathbb{R}^d$ and $n = o(d)$. Using a randomized rounding mechanism, we show that it suffices to consider only sparse $w$ as follows:

$$
\begin{aligned}
\text{variables} \quad & w_1, \ldots, w_d, A \\
\text{subject to} \quad & \forall i \in A : \sum_{j=1}^d Z_{i,j} w_j \geq 1 \\
& \forall j \in [d] : w_j \in \{0, 1\}, \\
& \sum_{j=1}^d w_j \leq \frac{2 \times 10^7}{\alpha} \\
& A \subset [n], |A| \geq 10^{-4} n
\end{aligned}
\tag{F5}
$$

The following claim shows that if (F4) is feasible then (F5) is also feasible.

**Claim D.6.** *If (F4) is feasible, then (F5) is also feasible.*

*Proof.* Let $w_1, \ldots, w_d$ and $A$ be the feasible solution to (F5). Set $q_j = \min(1, w_j/(\alpha\|w\|_1))$ for $j \in [d]$. Consider the following random assignment: set $W_j \sim \text{Ber}(q_j)$ independently for $j \in [d]$. We will show that with non-zero probability $W_j$'s satisfy (F5). Consider the following events:

$$
\mathcal{E}_1 := \left\{ \sum_{j=1}^d W_j \leq 2 \times 10^{-7}(1/\alpha) \right\}, \quad \text{and} \quad \mathcal{E}_2 := \left\{ |\{i : \sum_{j=1}^d Z_{i,j} W_j \geq 1\}| \geq 10^{-4} n \right\}
\tag{19}
$$

We will show that $\mathbb{P}\{cE_1\} \geq 1 - 5 \times 10^{-8}$ and $\mathcal{P}\{cE_2\} \geq 10^{-7}$. By a union bound, we will have that $\mathcal{E}_1 \cap \mathcal{E}_2$ has non-zero probability and thus (F5) is feasible.

Let $F := \{j \in [d] : W_j = 1\}$ be the set of coordinates where $W_j$ is non-zero. Then $\mathbb{E}[|F|] = \mathbb{E}[\sum_{j=1}^d W_j] = \sum_{j=1}^d q_j \leq 1/\alpha$. Thus with probability at least $1 - 5 \times 10^{-8}$, we have that the number of non-zero $W_j$'s is at most $\frac{2 \times 10^7}{\alpha}$. Equivalently, $\mathbb{P}\{\mathcal{E}_1\} \geq 1 - 5 \times 10^{-8}$.

We now focus on the second event $\mathcal{E}_2$. Let $S_1, \ldots, S_n$ be the subsets of $[d]$ such that $S_i = \{j \in [d] : Z_{i,j} = 1\}$, i.e., for each sample $i$, $S_i$ is the set of indices where the coordinates are large. Consider the random variables $R_1, \ldots, R_n$, where for $i \in [n]$, $R_i := \sum_{j=1}^d Z_{i,j} W_j = \sum_{j \in S_i} Z_{i,j} W_j$. (F5) requires that for at least $10^{-4}$ fraction of $i$'s, $R_i \geq 1$. Since $Z_{i,j}$'s are binary and fixed, we have that $R_i$ is distributed as Binomial random variable and is thus anti-concentrated.

**Fact D.7** (Anti-concentration of Binomial). *Let $X \sim Binomial(n, p)$ for some $n \in \mathbb{N}$ and $p \in [0, 1]$. Suppose $\mathbb{E}[X] \geq 1$. Then $\mathbb{P}\{X \geq 1\} \geq (1 - 1/e)$.*

*Proof.* Using the fact that $1 + x \leq e^x$ for all $x \in \mathbb{R}$, we get the following:
$$\mathbb{P}\{X \geq 1\} = 1 - \mathbb{P}\{X = 0\} = 1 - (1 - p)^n \geq 1 - (e^{-p})^n = 1 - e^{-np} \geq (1 - 1/e). \qquad \square$$

Consider a fixed $i \in A$. Then either there exists a $j \in S_i$ such that $q_j = 1$, or for all $j \in S_i$, $q_j < 1$. In the former case, we have that $R_i$ is at least one since $W_j = 1$.

In the latter setting, we have that $q_j = w_j/(\alpha \|w\|_1)$ for all $j \in S_i$, and thus $\mathbb{E}[R_i] = \sum_{j \in S_i} Z_{i,j} q_j = \sum_{j=1}^d Z_{i,j} w_j/(\alpha \|w\|_1) \geq 1$. Applying Fact D.7 to any such $i \in A$, we get that the probability of $R_i$ being positive is at least $1 - 1/e$. Let $A'$ be set of $i$'s such that $R_i \geq 1$, i.e., $A' = \{i : R_i \geq 1\}$. Thus combining the two cases above, we have the following:
$$\forall i \in A : \mathbb{P}\{i \in A'\} \geq 0.5. \tag{20}$$
Thus $\mathbb{E}[|A'|] \geq 0.5|A| \geq 5 \times 10^{-4}$. Since $|A'|$ lies in in $[0, n]$, applying Paley-Zygmund inequality to the random variable $|A'|$, we get the following:
$$\mathbb{P}\{|A'| \geq 10^{-4}n\} \geq \mathbb{P}\{|A'| \geq 0.2\,\mathbb{E}[|A'|]\} \geq 0.64\frac{(\mathbb{E}[|A'|])^2}{n^2} \geq 0.64 \times 25 \times 10^{-8} > 10^{-7}. \tag{21}$$
Equivalently, $\mathbb{P}\{\mathcal{E}_2\} \geq 10^{-7}$. This completes the proof. $\qquad \square$

Thus it suffices to show that, with high probability, (F5) is infeasible.

**Lemma D.8** (Infeasibility of (F5)). *Under the setting of Lemma D.1 and when $k > 10^{26}$, there exists a constant $c_1 > 0$ such that if $n \geq c_1(k^{1.5} \log d + \log(1/\tau))$, then with probability $1 - \tau$, (F5) is infeasible.*

*Proof.* First consider any fixed $w = (w_1, \ldots, w_d)$ such that $w_i \in \{0, 1\}$ and $\sum_{j=1}^d w_j \leq 2 \times 10^7 \cdot (1/\alpha)$.

Consider the integer-valued random variables $R_1, \ldots, R_n$ such that $R_i = \sum_{j=1}^d Z_{i,j} w_j$, and observe that $R_i$'s are i.i.d. random variables (since $X_i$'s are i.i.d. random variables). Thus (F5) requires that at least $10^{-4}\%$ of $R_i$'s are non-zero.

By the fourth moment bound on each coordinate, we have that $\mathbb{E}[Z_{i,j}] = \mathbb{P}\{X_{i,j} \geq 2\nu\sqrt{k}\} \leq 1/k^2$ for each $i$ and $j$. Thus the expectation of each $R_i$ is at most $\sum_{j=1}^d w_j \,\mathbb{E}[Z_{i,j}] \leq \sum_{j=1}^d w_j(1/k^2) \leq (2 \times 10^7)/(k^2\alpha) = (2 \times 10^7)/(k^2\alpha) = (6 \times 10^7)/\sqrt{k}$, which is less than $10^{-5}$ for $k$ large enough. By Markov's inequality, the probability that $\mathbb{P}\{R_1 \geq 1\} \leq 10^{-5}$.

Thus by Chernoff bound (since $R_i$'s are independent), with probability at least $1 - \exp(-c'n)$, the fraction of $R_i$'s that are non-zero is at most $5 \times 10^{-5}$. Hence, with the same probability, this particular choice of $w$ does not satisfy (F5). Since there are at most $d^{(2\times 10^7)/\alpha}$ such choices of $w$, applying a union bound, we get that (F5) is infeasible with probability at least $1 - \exp((2 \times 10^7)/\alpha) \cdot \log d - c'n)$. The failure probability is at most $\tau$ when $n \gtrsim \log(1/\tau) + k^{1.5} \log d$. This concludes the proof. $\qquad \square$

Since we assumed $k$ is large enough, Lemma D.8 is applicable. Lemma D.8 implies that, with high probability, the program (F5) is infeasible. Hence, with the same high probability, the programs (F4) and (F3) are also infeasible, and the programs (F1) and (F2) are feasible. This completes the proof. $\qquad \square$

### D.1 Lipschitz Argument

Let $h_{a,b}$ be as defined in Equation (1).

**Lemma D.9** (Lipschitzness of Truncation Under Coordinatewise Regularity)**.** *Let $\mu, \mu'$ be vectors in $\mathbb{R}^d$ and let $a \in \mathbb{R}_+$ be greater than 2. Let $S = \{x_1, \ldots, x_n\} \subseteq \mathbb{R}^d$ be the set of $n$ points. Suppose there exist a set $S_1 \subset [n]$ satisfying the following:*

$$|S_1| \geq 0.98n \ \ and \ \ \left\| \frac{1}{|S_1|} \sum_{i \in S_1} (h_{a,\mu}(x_i) - \mu')(h_{a,\mu}(x_i) - \mu')^\top \right\|_{\mathcal{X}_k} \leq r. \tag{22}$$

*Suppose also that, for some $\alpha \in (0,1)$, there exist a set $S_2 \subset [n]$ satisfying the following:*

$$|S_2| \geq 0.99n \ \ and \ \ \forall j \in [d] : \sum_{i \in S_2} \mathbb{I}_{|x_{i,j} - \mu_j| \geq a/2} \leq \alpha n. \tag{23}$$

*Then, we have the following: there exists a set $S_3 \subset [n]$ such that for all $b \in \mathbb{R}^d$ satisfying $\|b - \mu\|_\infty \leq a/2$ and $\|b - \tilde{\mu}\|_\infty \leq a$, we have that*

$$|S_3| \geq 0.97n \ \ and \ \ \left\| \frac{1}{|S_3|} \sum_{i \in S_3} (h_{a,b}(x_i) - \mu')(h_{a,b}(x_i) - \mu')^\top \right\|_{\mathcal{X}_k} \leq 1.1r + 5a\alpha k \|b - \mu\|_\infty. \tag{24}$$

*Proof.* We will take $S_3 = S_1 \cap S_2$, which directly implies that $|S_3| \geq 0.97n$. For any $M \in Chi_k$, since $xx^\top \bullet M \geq 0$, we have the following:

$$\left\langle M, \frac{1}{|S_3|} \sum_{i \in S_3} (h_{a,\mu}(x_i) - \mu')(h_{a,\mu}(x_i) - \mu')^\top \right\rangle \leq \left\langle M, \frac{1}{|S_3|} \sum_{i \in S_1} (h_{a,\mu}(x_i) - \mu')(h_{a,\mu}(x_i) - \mu')^\top \right\rangle$$

$$\leq \frac{1}{0.97} r.$$

Let $F(b)$ be the following matrix:

$$F(b) = \frac{1}{|S_3|} \sum_{i \in S_3} (h_{a,b}(x_i) - \mu')(h_{a,b}(x_i) - \mu')^\top$$

We will establish that $\|F(b) - F(\mu)\|_{\chi_k} \leq 5a\alpha k \|b - \mu\|_\infty$, which establishes the lemma statement by the triangle inequality. In order to do that, we will show that $\|F(b) - F(\mu)\|_\infty \leq 5a\alpha \|b - \mu\|_\infty$ and then use Lemma A.6.

Consider an arbitrary $(j, \ell)$-entry of these matrices. By abusing notation, when $x$ and $y$ are scalar, we use $h_{a,y}(x)$ to be the function from $\mathbb{R} \to \mathbb{R}$ defined analogously to Equation (1). Let $g(\cdot, \cdot)$ be the following function that is equal to the $(j, \ell)$ entry of the matrix $F(b)$, which is explicitly

$$g(b_j, b_\ell) = \frac{1}{|S_3|} \sum_{i \in S_3} (h_{a,b_j}(x_j) - \mu'_j)(h_{a,b_\ell}(x_\ell) - \mu'_\ell).$$

We will show that $g(\cdot, \cdot)$ is locally Lipschitz in its arguments. Consider a particular $i \in S_3$ and define the following:

$$g_i(b_j, b_\ell) = (h_{a,b_j}(x_{i,j}) - \mu'_j)(h_{a,b_\ell}(x_{i,\ell}) - \mu'_\ell)$$

Then, we can bound the difference for each sample by

$$|g_i(b_j, b_\ell) - g_i(\mu_j, \mu_\ell)|$$
$$= |(h_{a,b_j}(x_{i,j}) - \mu'_j)(h_{a,b_\ell}(x_{i,\ell}) - \mu'_\ell) - (h_{a,\mu_j}(x_{i,j}) - \mu'_j)(h_{a,\mu_\ell}(x_{i,\ell}) - \mu'_\ell)|$$
$$\leq |(h_{a,b_j}(x_{i,j}) - h_{a,\mu_j}(x_{i,j}))(h_{a,b_\ell}(x_{i,\ell}) - \mu'_\ell)| + |(h_{a,\mu_j}(x_{i,j}) - \mu'_j)(h_{a,b_\ell}(x_{i,\ell}) - h_{a,\mu_\ell}(x_{i,\ell}))|$$
$$\leq (a + \|b - \mu'\|_\infty) \cdot \|b - \mu\|_\infty \left( \mathbb{1}_{|x_{i,j} - \mu_j| \geq a - \|b - \mu\|_\infty} + \mathbb{1}_{|x_{i,\ell} - \mu_\ell| \geq \|b - \mu\|_\infty} \right)$$
$$\leq (a + \|b - \mu'\|_\infty) \cdot \|b - \mu\|_\infty \left( \mathbb{1}_{|x_{i,j} - \mu_j| \geq a/2} + \mathbb{1}_{|x_{i,\ell} - \mu_\ell| \geq a/2} \right)$$
$$\leq 2a \cdot \|b - \mu\|_\infty \left( \mathbb{1}_{|x_{i,j} - \mu_j| \geq a/2} + \mathbb{1}_{|x_{i,\ell} - \mu_\ell| \geq a/2} \right),$$

where we use that $|h_{a,y}(x) - h_{a,z}(x)| \leq |y - z|$, $|h_{a,y}(x) - z| \leq |a + y - z|$, and $|h_{a,y}(x) - h_{a,z}(x)|$ is non-zero only if $|x - y| \geq a - |y - z|$.

Combined with assumption Equation (23), this implies that

$$|g(b_j, b_\ell) - g(\mu_j, \mu_\ell)| \leq 2a \cdot \|b - \mu\|_\infty \cdot \frac{2\alpha}{0.97} \leq 5a\alpha\|b - \mu\|_\infty.$$

By Lemma A.6, we have the following:

$$\|F(b) - F(\mu)\|_{\mathcal{X}_k} \leq k\|F(b) - F(\mu)\|_\infty \leq 5a\alpha k \cdot \|b - \mu\|_\infty.$$

$\square$

## D.2 Main Theorem

**Lemma D.10.** *Let $S$ be a set of $n$ i.i.d. data points from a distribution $P$ over $\mathbb{R}^d$. Let the mean of $P$ be $\mu$, and covariance $\Sigma$ such that $\|\Sigma\|_{\mathcal{X}_k} \leq \sigma^2$, and for all $i \in [d]$, $\mathbb{E}[X_i^4] \leq O(\sigma^4)$. Suppose $n = \Omega(k^2 \log d + \log(1/\tau))$. Let $a = \sigma\sqrt{k}$. With probability $1 - \tau$ over $S$, there exists a subset $S' \subset S$ with $|S'| \geq 0.95n$ such that for any $b$ satisfying $\|b - \mu\|_\infty = O(\sigma)$, we have $h_{a,b}(S')$ is $(0.01, O(1), k)$-stable with respect to some $\mu'$ and $\sigma$ with $\|\mu' - \mu\|_\infty \leq O(\sigma/\sqrt{k})$.*

*Proof.* Let $P'$ be distribution of $h_{a,\mu}(P)$ and let $\mu'$ and $\Sigma'$ be the mean and covariance of $P'$. This will be the $\mu'$ in the lemma statement. By Lemma 3.1, we get that (i) $\|\mu' - \mu\|_\infty \leq \sigma/\sqrt{k}$, , (ii) $\|\Sigma - \Sigma'\|_{\mathcal{X}_k} \leq O(\sigma^2)$, and (iii) $P'$ is supported on the set $\{x : \|x - \mu\|_\infty \leq a\}$.

Applying Theorem C.1 to $P'$ states that with probability at least $1 - \tau$, there exists a subset $S_1 \subset S$ with $|S_1| \geq 0.98n$ such that

$$\left\| \frac{1}{|S_1|} \sum_{i \in S_1} (h_{a,\mu}(x_i) - \mu')(h_{a,\mu}(x_i) - \mu')^\top \right\|_{\mathcal{X}_k} \leq O(\sigma^2). \tag{25}$$

By applying Lemma D.1 to $P - \mu$, with probability at least $1 - \tau$, there exists a subset $S_2 \subset S$ with $|S_2| \geq 0.99n$ such that

$$\forall j \in [d] : \sum_{i \in S_2} \mathbb{I}_{|x_{i,j} - \mu_j| \geq a/2} \leq O(k^{-1.5})n. \tag{26}$$

We can then apply Lemma D.9 to show that, conditioned on the above events, there exists a subset $S_3 \subset S$ with $|S_3| \geq 0.97n$ such that for all $b$ such that $\|b - \mu\|_\infty \leq O(\sigma)$ and $\|b - \mu'\|_\infty \leq O(\sigma)$ (the latter holds by the triangle inequality for all $b$ such that $\|b - \mu\|_\infty \leq O(\sigma)$), we have that

$$\left\| \frac{1}{|S_3|} \sum_{i \in S_3} (h_{a,b}(x_i) - \mu')(h_{a,b}(x_i) - \mu')^\top \right\|_{\mathcal{X}_k} \leq O(\sigma^2) + O(ak^{-1.5}k\|b - \mu\|_\infty) \leq O(\sigma^2). \tag{27}$$

By Proposition A.1, this implies $S_3$ contains a set $S'$ satisfying the following: (i) $|S'| \geq 0.95n$ is $(0.1, O(1), k)$-stable with respect to $\mu'$ and $\sigma$. Thus, we choose $S'$ in the lemma statement to be this set.

Taking a union bound, all the above events fail with probability at most $O(\tau)$. Reparameterizing yields the lemma statement.

$\square$

**Theorem D.11.** *Let $S$ be a set of $n$ i.i.d. data points from a distribution $P$ over $\mathbb{R}^d$, and let $T$ be a $0.01$-corruption of $S$. Let $\tilde{\mu}$ be the coordinate-wise median-of-means estimate computed from set $T$. Let the mean of $P$ be $\mu$, and covariance $\Sigma$ such that $\|\Sigma\|_{\mathcal{X}_k} \leq \sigma^2$, and for all $i \in [d]$, $\mathbb{E}[X_i^4] \leq O(\sigma^4)$. Suppose $n = \Omega(k^2 \log d + \log(1/\tau))$. Let $a = \sigma\sqrt{k}$. With probability $1 - \tau$ over $S$, for all $T$ we have that there exists a subset $S' \subset T$ with $|S'| \geq 0.95n$ such that $h_{a,\tilde{\mu}}(S')$ is $(0.01, O(1), k)$-stable with respect to some $\mu'$ and $\sigma$ with $\|\mu' - \mu\|_\infty \leq O(\sigma/\sqrt{k})$.*

*Proof.* By Fact 2.1, we know that with probability at least $1 - \tau$, we have $\|\tilde{\mu} - \mu\|_\infty \leq O(\sigma)O(1 + (\log(d/\tau))/n) = O(\sigma)$ by the assumption that $n$ is sufficiently large.

Thus, we use $\tilde{\mu}$ as "$b$" in Lemma D.10 to yield the stability guarantee in the theorem statement.

The total failure probability is at most $2\tau$, and reparameterizing yields the theorem statement. $\square$

In Section 4, we use Theorem D.11 as the main technical ingredient to prove Theorem D.12, which in turn implies Theorem 1.3.

**Theorem D.12** (Stronger version of our result). *Let $\epsilon \in (0, \epsilon_0)$ for small constant $\epsilon_0 > 0$. Let $P$ be a multivariate distribution over $\mathbb{R}^d$, where the mean and covariance of $P$ are $\mu$ and $\Sigma$ respectively. Suppose $\|\Sigma\|_{\mathcal{X}_k} \leq 1$ and further suppose that for all $j \in [d]$, $\mathbb{E}[(X_j - \mu_j)^4] = O(1)$. Then, there is an algorithm such that, on input (i) the corruption parameter $\epsilon$, (ii) the failure probability $\tau$, (iii) the sparsity parameter $k$, and (iv) $T$, an $\epsilon$-corrupted set of $n \gg (k^2 \log d + \log(1/\tau))/\epsilon)$ i.i.d. samples from $P$, it outputs $\hat{\mu}$ satisfying $\|\hat{\mu} - \mu\|_{2,k} = O(\sqrt{\epsilon})$ with probability $1 - \tau$ in $\mathrm{poly}(n, d)$ time.*

## E  Information-Theoretic Error

Let $\mathcal{D}_k$ be the family of all distributions over $\mathbb{R}^d$ that satisfy the following:

1. For every $D \in \mathcal{D}_k$, the mean of $D$ is $k$-sparse,

2. For every $D$ in $\mathcal{D}_k$ the covariance of $\mathcal{D}$ is upper bounded by $I$ in spectral norm, and

3. For every $D \in \mathcal{D}_k$ we have that $\mathbb{E}[(X_i - \mathbb{E}[X_i])^4] = O(1)$, where $X = (X_1, \ldots, X_d) \sim D$.

**Lemma E.1.** *Let $k \geq 1/\sqrt{\epsilon}$. Then there exist two distributions in $D_1$ and $D_2$ in $\mathcal{D}_k$ such that the following hold: (i) $d_{TV}(D_1, D_2) = \epsilon$, and (ii) The means of $D_1$ and $D_2$ are separated by $\Omega(\sqrt{\epsilon})$ in $\ell_{2,k}$-norm.*

Before giving the proof of Lemma E.1, we remark that the assumption of $k \geq 1/\sqrt{\epsilon}$ is mild. First, the assumption is independent of the ambient dimensionality $d$—the most challenging parameter regime in algorithmic robust statistics is when we fix a small $\epsilon$ and then take the dimensionality $d$ to $\infty$. Second, the typical interesting sparsity regime is when $k$ is super-constant but grows very slowly in $d$, say, logarithmically. The assumption that $k \geq 1/\sqrt{\epsilon}$ applies readily to the above regime.

*Proof.* Let $D_1$ be the distribution that places all of its mass at origin, i.e., $(0, \ldots, 0)$. Let $D_2$ be the distribution that places $(1 - \epsilon)$ probability mass at origin and places $\epsilon$ probability mass at $y$, where the first $k$-coordinates of $y$ are $\alpha$ and the remaining $d - k$ coordinates are 0.

It is easy to see that the total variation distance between $D_1$ and $D_2$ is $\epsilon$, and that $D_1 \in \mathcal{D}_k$. We will now show that $D_2 \in \mathcal{D}_k$ for a suitable value of $\alpha$.

1. First the mean of $D_2$ is $\epsilon y$, which is $k$-sparse by construction.

2. We have that the covariance of $D_2$ is $\epsilon y y^\top - \epsilon^2 y y^\top = \epsilon(1 - \epsilon)y y^\top \preceq \epsilon y y^\top$, which is upper bounded by 1 in spectral norm if $\|y\|_2 \leq 1/\sqrt{\epsilon}$. Since $\|y\|_2 = \sqrt{k}\alpha$, we want that $\alpha \leq 1/\sqrt{k\epsilon}$.

3. Finally, let $X \sim D_2$. For every $i > k$, we have that $\mathbb{E}[(X_i - \mathbb{E}[X_i])^4] = 0$. For $i \in [k]$, $\mathbb{E}[(X_i - \mathbb{E}[X_i])^4] = \mathbb{E}[(X_i - \epsilon\alpha)^4] \leq 8(\mathbb{E}[X_i^4 + \epsilon^4\alpha^4]) = 8(\epsilon\alpha^4 + \epsilon^4\alpha^4) \leq 16\epsilon\alpha^4$, which is less than 16, if $\alpha \leq \epsilon^{-1/4}$.

Thus the above construction goes through as long as $\alpha \leq \min(1/\sqrt{k\epsilon}, \epsilon^{-1/4})$. When $k \geq 1/\sqrt{\epsilon}$, it suffices that $\alpha = 1/\sqrt{k\epsilon}$. Finally, we note that the difference in means of $D_1$ and $D_2$ is $\epsilon\|y\|_2 = \epsilon\sqrt{k}\alpha = \sqrt{\epsilon}$ for the chosen value of $\alpha$. $\square$