# OpenReview forum: "Outlier-Robust Sparse Mean Estimation for Heavy-Tailed Distributions"
_NeurIPS.cc/2022/Conference — NeurIPS 2022 Accept_

### Official Review · Reviewer_VSiz · 2022-07-06

**Rating:** 7
**Confidence:** 4
**Soundness:** 4 excellent
**Presentation:** 4 excellent
**Contribution:** 3 good

**Summary:**

Algorithmic robust statistics is concerned with building efficient algorithms for statistical estimation problems when the observed data contains adversarial or heavy-tailed noise. For the canonical problem of mean estimation, a single grossly corrupted data point can completely invalidate the performance of the sample average as a natural estimate of the population mean. In the problem of heavy-tailed mean estimation under the strong contamination model, one observes $n$ data points generated in the following manner:

1. First $n$ ``good'' data points are generated from the true underlying heavy-tailed distribution $D$. Note that assumptions on $D$ are usually very mild -- often only that its variance exists.
2. An adversary then inspects the generated samples and corrupts an arbitrary set of $\epsilon n$ points in a completely unconstrained manner.

Given a target failure probability $\delta$ and corrupted dataset  $X = \left\\{x_i\right\\}_{i = 1}^n \subset \mathbb{R}^d$, the goal is to optimally recover the mean of $D$ generating the good data points. Here, optimality is measured in terms of the relevant parameters of interest $n, d, \delta, \epsilon$. With no further assumptions, computationally efficient algorithms have been designed achieving the information theoretically optimal rate of:
\begin{equation*}
    O \left(\sqrt{\frac{d}{n}} + \sqrt{\frac{\log (1 / \delta)}{n}} + \sqrt{\epsilon}\right).
\end{equation*}
Note that this rate is optimal upto constants -- in fact, a family of Gaussians witness the terms depending on $d$ and $\delta$ despite making no assumptions on the higher-order moments of $D$.

The paper considers the sparse setting where the mean, $\mu$, of $D$ is $k$-sparse. In typical sparse estimation problems, this usually allows an improvement of the sample complexity to depending polynomially only on $k$ and poly-logarithmically on $d$. In this paper, the authors construct an algorithm for sparse heavy-tailed mean-estimation achieving optimal recovery when $n \gg (k^2 \log (d) + \log (1 / \tau)) / \epsilon$ which is optimal under the conjectured statistical computational gap (information theoretic recovery is possible with $(k \log (d) + \log (1 / \delta)) / \epsilon$ samples). The paper imposes a mild assumption on the higher order moments and assumes additionally that $\mathbb{E} [(X_i - \mu_i)^4] = O(1)$.

Technically, the paper uses a well-established stability based framework for robust estimation with some minor modifications. Specifically, the data points are truncated in the $\ell_\infty$ norm (using a coarse estimate of the true mean using the coordinate-wise median-of-means estimate) to remove spikes along sparse directions and utilizing a standard median-of-means pre-processing technique. Having established these stability conditions, the algorithm follows largely from the prior work.

In the dense setting, the goal is to establish a set of well-spread weights $w$ such the empirical covariance of the data points weighted by $w$, $\Sigma_w$, is small in spectral norm. Intuitively, $w$ would ideally be the uniform distribution over the good data points. Formally, one requires:

$$
    \min_w \lVert \Sigma_w \rVert = \min_w \max_{M \succcurlyeq 0, \mathrm{Tr} M = 1} \langle \Sigma_w, M \rangle = \max_{M \succcurlyeq 0, \mathrm{Tr} M = 1}  \min_{w} \langle \Sigma_w, M \rangle = O(1)
$$

where the inversion of the min and max is guaranteed by von-Neumann's min-max inequality. The last step is crucial since it suffices to design a set of weights depending on the ``direction'' $M$. Through a clever rounding argument from [DL 22], it suffices to only consider the case where $M$ is rank $1$ up to appropriate adjustments in constants. Therefore, one only requires $d$ data points to effectively cover the set of plausible directions as opposed to the $d^2$ sized cover of $M$.

In the sparse setting, it suffices to only search for sparse directions leading to the following modification of the above equation:

$$
    \min_{w} \lVert \Sigma_w \rVert = \min_{w} \max_{M \succcurlyeq 0, \mathrm{Tr} M = 1, \lVert M \rVert_1 \leq k} \langle \Sigma_w, M \rangle = \max_{M \succcurlyeq 0, \mathrm{Tr} M = 1, \lVert M \rVert_1 \leq k}  \min_{w} \langle \Sigma_w, M \rangle = O(1)
$$

To prove the stability condition, the authors proceed through a similar rounding argument as [DL 22] with the additional difficulty that it is not clear how to round such a $M$ to a sparse vector/matrix. In the dense case, it suffices to draw a zero-mean Gaussian with covariance $M$. Inspired by [Li 18] (where a decomposition of such a matrix into a sum of sparse matrices is carried out in Lemma 3.4.4), the authors design a careful probabilistic decomposition of $M$ to carry out this approach.

[DL 22] J. Depersin and G. Lecué. Robust sub-Gaussian estimation of a mean vector in nearly linear time.

[Li 18] J. Li. Principled Approaches to Robust Machine Learning and Beyond.

**** POST REBUTTAL UPDATE ****

I acknowledge the authors' response and will retain my current evaluation.

**Questions:**

My main question to the authors is the requirement of higher-order moments. In the example provided in the paper, they construct a distribution centered at $0$ with identity covariance that when $\ell_\infty$ truncated at $\sqrt{k}$, whose covarince increases in spectral norm to $O(d / k)$. However, this is along an extremely dense direction oriented along the all 1s vector. It still remains the case that if one were only looking along sparse directions, the variance would remain small. In this case, it is not obvious to me that higher order moments are required. It would be immensely helpful if the authors could comment further on the necessity of these assumptions.


**Limitations:**

Yes

**Strengths And Weaknesses:**

This paper makes significant progress the problem of building robust estimators for sparse mean estimation under heavy tailed and adversarial noise. Prior results only held for dense setting or under extremely strong distributional assumptions such as sub-gaussianity. However, one downside to the paper is the lack of technical novelty the design and analysis of the estimator. The algorithm follows naturally from well-established frameworks and its subsequent analysis is based on a rounding approach from prior work. However, the decomposition of sparse matrices for the rounding step to go through is novel and interesting.

---

> ### Author Response · Authors · 2022-08-01
> **Authors' response to Reviewer VSiz**
>
> Thank you for your positive evaluation of our paper, especially of our non-spectral decomposition of sparse positive semi-definite matrices, and for your question.
>
> + (Bound on the axis-wise fourth-moment) To answer your question: the example in Appendix A.4 demonstrates a distribution where truncation blows up the spectral norm (indeed, in a dense direction). This is not directly used to justify the higher moment assumption, but rather, to justify that instead of trying to preserve the spectral norm, we should rather aim for a weaker result, namely that the $\mathcal{X}_k$ norm of the covariance is preserved after truncation. (The $\mathcal{X}_k$ norm is a convex relaxation of the “spectral norm restricted to $k$-sparse directions”.) It is true that after truncation, the variance in each axis direction can only decrease, but for slightly denser directions, the variance might nonetheless blow up. The axis-only 4th moment assumption is then imposed to prevent this scenario, and we are able to show that the $\mathcal{X}_k$ norm in general is preserved under truncation. $\qquad\qquad\qquad\qquad\qquad\qquad\qquad\qquad\qquad\qquad\qquad\qquad\qquad\qquad\qquad\qquad\qquad\qquad\qquad\qquad\qquad\qquad\qquad\qquad$
>
>    The axis-only 4th moment assumption also appears in another spot in the rounding argument (see Lines 386-391 in the proof of Theorem 5.1, more precisely, Lemma C.4). We do conjecture that ultimately, the assumption should not be necessary, that the same error rate should be attainable with a poly-time algorithm under only the bounded covariance assumption. However, the challenges we encountered in using the filtering-based approach suggest that a drastically different algorithmic approach might potentially be needed.
>
> ---
>
> + *“One downside to the paper is the lack of technical novelty the design and analysis of the estimator”*
>
>   While we do not claim any novelty in the design of the estimator (algorithm), we do believe that the analysis of our algorithm is novel and technically challenging in a number of ways. Here, we wish to re-emphasize these aspects of our analysis.
>
>    + (Novel rounding mechanism) We appreciate the reviewer’s positive assessment of our non-spectral decomposition. We want to further emphasize (see our overall response as well) that the decomposition is designed to tailor to the (unknown) inlier distribution.
>
>    + We want to also highlight a different appealing technical aspect of our analysis arising from handling strong contamination instead of just Huber (aka oblivious) contamination.
>         + Our algorithm first computes a preliminary mean estimate, uses it to perform truncation, before recomputing the final mean estimate. In the non-robust setting, such an algorithm template is relatively straightforward to analyze: use a small fraction of samples for the preliminary estimate, and use the rest of the samples for the final (main) estimation. The two sets are independent, and error probabilities can be easily bounded via a union bound.
>
>         + However, the analysis becomes far more subtle in the setting with adversarial contamination. Recall that, in the strong contamination model, the algorithm inspects all the inlier samples before replacing an arbitrary small fraction of the samples with arbitrary outliers. If we attempted partitioning the samples (as in the non-robust setting), we would quickly run into statistical dependency issues. Concretely, it is *no longer true* that the two sets of corrupted samples are generated independently, as the corruptions in one set have potential dependencies on the samples from the other set.
>
>         + In our analysis, we do not partition the samples to do preliminary and final estimation separately, and instead reuse the entire set of samples, given the partitioning still introduces dependencies anyway. We analyze the two-step approach directly by establishing that our pre-processing step is Lipschitz except with inverse exponential probability. Our arguments rely on an intricate argument involving LP duality (see Section D in Appendix), which may also be of independent interest.
>
>         + Note also that the above dependency challenge does not arise if we had just assumed the Huber contamination model. Importantly, with the argument in Appendix D, we are also able to handle the strong contamination model.

---

### Official Review · Reviewer_kKU6 · 2022-07-09

**Rating:** 3
**Confidence:** 4
**Soundness:** 2 fair
**Presentation:** 3 good
**Contribution:** 1 poor

**Summary:**

The paper study robust mean estimation in a sparse high-dimensional setting. Specifically, the goal is to exhibit an estimator that verify the following properties: the estimator has a sub-gaussian concentration  around the true mean when the data are heavy-tailed, the error due to a proportion epsilon of corrupted data can be controlled and is of order $O(\sqrt(\epsilon))$.

The authors use the $\ell_2$ norm of the $k$ largest coordinates of a vector in magnitude to define a sparse distance and they show how to construct an estimator verifying the aforementioned properties with respect to this distance. Thanks to this sparse notion of distance they replace the usual dimension-dependent error by an error that depends on $k$ which can be a lot larger when the data are truly sparse.

**Questions:**


- The estimator does not seem computable, which contrast with the author's claim that it is. The theorems provided in the article guarantee a polynomial complexity, however this does not guarantee that the estimator is computable in practice. Moreover, the choice of constants given in Section C.1 does not seem to be practical. Finally, no experiment were given with the article which seems to indicate that it is not usable in practice. Could the authors comment on that ?

- Would the analysis still work with only two finite moments  ? Is a fourth finite moment something that is needed for the algorithm to work ?

- The algorithm for now needs to know $\epsilon$. Is there a hope to give an algorithm that does not need the knowledge of $\epsilon$ ? For example the Tukey median does not need the knowledge of $\epsilon$ but it is not computable.

- The constants are hidden even though it can be very important to have tight constants in front of the leading term (See Catoni "Challenging the empirical mean and empirical variance: a deviation study" (2012) and the more recent work of S.Minsker "U-statistics of growing order and sub-Gaussian mean estimators with sharp constants" (2022) investigating this problem in dimension 1). If the constants are very far from being tight, it should be said in the article.

- Even though the authors claim that $O(\sqrt{\varepsilon})$ is optimal in your setting, this does not seem to be true. When the data have a finite fourth moment as in Theorem, $O(\varepsilon^{3/4})$ and more generally when the data have $q$ finite moments, the optimal error is $O(\varepsilon^{1-1/q})$ (see S. Minsker "Uniform bounds for robust mean estimators" for example). This discrepancy seems to be due to the requirement of $k \ge 1/\sqrt{\varepsilon}$ in Lemma E.1. When $\varepsilon$ goes to $0$, this requirement is very strong. On the other hand, when $\varepsilon$ is larger than some constant $\varepsilon_0$, then we can say that $\sqrt{\varepsilon}\le \varepsilon/\varepsilon_0$ and hence $O(\sqrt{\varepsilon})=O(\varepsilon)$ which makes me think that the interesting case is when $\varepsilon$ goes to $0$. Could the authors comment about that ?


**Limitations:**

No potential societal impact.

**Strengths And Weaknesses:**

Strengths: This is mostly clear and states the assumptions and results properly. In general finding an estimator exhibiting optimal sub-gaussian rate and optimal corruption error is hard and this is the first algorithm I know that manage this in the sparse setting.

Weaknesses: The paper claims that the algorithm used can be used in practice but I have doubts about that (more details in the questions). Some of the constants are hidden and it would be interesting the magnitude of these constant clearly. The paper is not very original and the theoretical results are not very surprising for an untractable algorithm.

---

> ### Author Response · Authors · 2022-08-01
> **Authors' response to Reviewer kKU6**
>
> Thank you for the detailed feedback. Here we address the questions raised in the review.
>
> 1. (Practical implementation of the algorithm) Indeed, our result is theoretical, and we do not claim that the current result works well in practice. We agree that improving the (practical) runtime is an important issue for future work. Nonetheless, this is the first polynomial-time algorithm for the problem with polynomial sample complexity, which *additionally* achieves both the tight sample complexity (up to conjectured computational-statistical tradeoff) and sub-Gaussian rate. We view this as a significant theoretical contribution.
> ---
> 2. (Necessity of the 4-th moment bound) The analysis does not work when there are only two finite moments, hence our mild extra assumptions. We use the mild fourth-moment bound in two places: the preservation of the covariance matrix in
> $\mathcal{X}_{k}$ norm after truncation, as well as Lemma C.4 (see Line 392) in the rounding argument. We emphasize though that our fourth moment assumptions are *only* along the axis directions, and imposes no restrictions on the exponentially many other directions in the $d$-dimensional space. (See also point 5 below, and our response to reviewer **VSiz**)
> ---
> 3. (Knowledge of $\epsilon$) The knowledge of $\epsilon$ is a common assumption within the line of work in algorithmic robust statistics. Lepskii’s method [L91, Bir01] can be used to reduce the unknown-$\epsilon$ case to the known-$\epsilon$ case in a black box manner (see, for example, [DL22, JOR22] for recent implementations of Lepskii’s method) with at most logarithmic increase in sample complexity and runtime. We will add this discussion in the next version of the paper.
> ---
> 4. (Size of the involved constants) We agree that constants are important in practice, and we are aware of and did cite (see Line 225) the line of work spanning Catoni’s estimator to the recent results of Lee and Valiant (2021) and Minsker (2022), settling the multiplicative constant in the 1D estimation problem. As we mentioned in the paper, our contribution is a theoretical result and we made no attempts to optimize any of the constants in the analysis (see Line 315). The constants we use in the current analysis are indeed far from optimal. We will state this more upfront and emphasize the theoretical nature of our work.
> ---
> 5. (Optimal error under bounded 4-th moments)
>     + We stress again that our fourth moment assumption is *only* along the axis directions. The reviewer’s claim regarding the optimal error being $\epsilon^{3/4}$ holds only under the much stronger assumption when the 4th moments are bounded in *every* $k$-sparse direction.
>
>     + Formally, the $\sqrt{\epsilon}$ information-theoretic lower bound is stated in Lemma E.1. We believe that the assumption $k \ge 1/\sqrt{\epsilon}$ is very mild: the most challenging (and practically relevant) parameter regime in algorithmic robust statistics is when we fix a small $\epsilon$ and then take the dimensionality $d$ to $\infty$. Typically, the most interesting sparsity regime is when $k$ is super-constant but grows very slowly in $d$ (say, logarithmically). The construction in Lemma E.1 applies readily to this parameter regime.
>
>     + We would also like to emphasize again that even under the much stronger assumption of bounded $q$-th moment for every direction for any constant $q$, none of the previous poly-time algorithms gives dimension-independent error with $\text{poly}(k (\log d)/\epsilon)$ samples (in fact, as shown in Example 1, prior algorithms provably fail even under this stronger assumption).
>
>     + Even in the dense setting (no sparsity) and bounded $q$-th moment for every direction for any  constant $q>2$, there is no known polynomial time algorithm that achieves error $o(\sqrt{\epsilon})$ with $\text{poly}(d/\epsilon)$ samples, even though $\epsilon^{1-1/q}$ is statistically possible. All of the known polynomial time algorithms that achieve error $o(\sqrt{\epsilon})$ impose very strong assumptions: either identity covariance, or sum-of-squares certifiability of moments. The latter approach further requires significantly more runtime and samples via solving SoS SDPs.
> ---
> **References**:
> + [L91] O. V. Lepskii. On a problem of adaptive estimation in Gaussian white noise. Theory of Probability & Its Applications, 35(3):454–466, 1991.
> + [Bir01] L. Birgé. An alternative point of view on Lepski’s method. Lecture Notes-Monograph Series, pages 113–133, 2001.
> + [DL22] J. Depersin and G. Lecué. Robust sub-Gaussian estimation of a mean vector in nearly linear time. The Annals of Statistics, 50(1):511–536, 2022.
> + [JOR22] A. Jain, A. Orlitsky, and V. Ravindrakumar, Robust estimation algorithms don’t need to know the corruption level, 2022.

---

> > ### Comment · Reviewer_kKU6 · 2022-08-04
> > **Remark regarding comparison with literature**
> >
> > * Regarding the comparison with other works, you say "the robust+sparse+heavy-tailed mean estimation problem previously had no polynomial-time estimators". I beleive that this is true, but even though the main contribution of your article is the sparse setting, you could still benefit to comparison with other algorithms (for some even in linear-time) in the case of robust to heavy-tail distribution.
> > In particular because your results basically say that in the sparse setting we can replace the dimension with the "effective dimension" (here $k$) and then have stronger guarantees than in non-sparse setting. I am also thinking about the speed $O(\sqrt{\epsilon})$ which you could explain more (I understand your answer about this, but this could be clarified in the article).
> >
> > * Regarding the knowledge on $\epsilon$: there are some algorithms that do not suppose this knowledge like M-estimators and Median-type algorithms. And also Lepski's method is often computationally intractable and from my experience it can be inefficient (with respect to constants) but as neither efficiency nor practical tractability really apply to your article I understand that Lepski's method can be an argument in your case.
> >
> > * About the regime of $\epsilon$ for which the results are interesting, remark that $\sqrt{\epsilon}$ is the order of magnitude known to be a lower bound for $\epsilon$ sufficiently small. For $\epsilon$ not close to zero, it is known from asymptotic arguments in robust statistics that the bias due to corruption should be of order $F^{-1}(\frac{1}{2(1-\epsilon)})$ where $F$ is the cdf of the inlier distribution (see for instance the book p72 of Robust Statistics by Huber and Ronchetti). Hence from my understanding I would think that when $\epsilon$ is a constant and is not close to $0$, then $\sqrt{\epsilon}$ may not be optimal. This depends strongly on the distribution.

---

> > > ### Author Response · Authors · 2022-08-06
> > > **Response to the reviewer**
> > >
> > > We respond to the reviewers’ additional comments below:
> > > + (Comparisons with other work)
> > >      + (Theoretical comparisons) We have already included comparisons in the paper saying that these prior algorithms require at least $d$ samples, which might be exponentially larger than $k$ in the regime when $k = \text{polylog}(d)$ (which is a standard interesting sparsity regime). In other words, prior algorithms for the dense setting may require exponentially more samples compared to what our algorithm requires.
> > >
> > >      + (Experimental comparisons) As our paper is theoretical in nature, there is little insight to be gained from experimentally comparing it with algorithms that either *provably fail* or *take exponentially more samples* in the sparsity to run.
> > >  + (Lepski’s method) We would like to note that the theoretical guarantee of Lepski’s method leads to at most a loss of a factor of 3 in the error, see, e.g., [JOR22, Lemma 3].
> > >  + (Lower bound) The subsequent points raised by the reviewer are also factually incorrect, which we explain as follows:
> > >
> > >     + “*that $\sqrt{\epsilon}$ is the order of magnitude known to be a lower bound for ϵ sufficiently small*”
> > >     + "*it is known from asymptotic arguments in robust statistics that the bias due to corruption should be of order $F^{−1}(1/2(1−ϵ))$*"
> > >
> > >    The $\sqrt{\epsilon}$ lower bound in Lemma E.1 holds for every $\epsilon \in (0,1/2)$. The result reported by the reviewer is for the (univariate) Gaussian distribution. More generally, the reported result characterizes the maximum asymptotic bias only on those (univariate) distributions that are both symmetric and unimodal, which is achieved by the sample median. That particular result does not apply to the heavy-tailed setting: it is well-known that the sample median is NOT a good estimator of the mean for general distributions due to large bias.
> > >
> > >    To reiterate, the lower bound for heavy-tailed distributions is $\Omega(\sqrt{\epsilon})$, which holds for every $\epsilon \in (0,1/2)$. Results of this form are well-known in the literature, see, e.g., [KS17, Lemma 7.1] among many others.
> > > ---
> > > [KS17] P. Kothari, D. Steurer. 2017. Outlier-robust moment-estimation via sum-of-squares. arXiv:1711.11581

---

> > > > ### Comment · Reviewer_kKU6 · 2022-08-06
> > > > **Rate with respect to $\epsilon$**
> > > >
> > > > The fact that $\sqrt{\epsilon}$ is tight only around $0$ is not only a univariate fact. In your setting where you take $\epsilon$ to be bounded away from $0$, say $\epsilon \ge \epsilon_0$, we have that $\sqrt{\epsilon}\le \epsilon/sqrt{\epsilon_0}=O(\epsilon)$. Hence, having tight speed of $\sqrt{\epsilon}$ up to a multiplicative constant is not very interesting when $\epsilon$ is not close to $0$. And it is not clear that the median is not good in high dimension, Tukey median for instance is known to be efficient.

---

> > > > > ### Author Response · Authors · 2022-08-08
> > > > > **Response**
> > > > >
> > > > > + We are baffled by the first comment made by the reviewer. (Of course, asymptotic notation loses its meaning if the argument is a fixed constant.) In the recent literature on algorithmic high-dimensional robust statistics, the ultimate goal for bounded covariance distributions is to achieve error O(\sqrt{\epsilon}) – i.e., the correct function of $\epsilon$ and independent of the dimension –  in the presence of an $\epsilon$-fraction of corrupted data. Given these facts, we reject the view that such a bound is “*not very interesting*”.
> > > > >
> > > > > +  (Tukey median) The reviewer writes “*Tukey median for instance is known to be efficient*”.
> > > > > Any reasonable interpretation of this statement is factually incorrect.
> > > > >
> > > > >     + First, the Tukey median is known to be NP-hard to compute in general, i.e., all known algorithms to compute it (even approximately) require exponential time (in the dimension). That is, the Tukey median does not yield a computationally efficient algorithm.
> > > > >     + Second, the Tukey median is not statistically efficient (even in one dimension!), if the underlying distribution is not symmetric. Specifically, the maximum $\epsilon$ it can tolerate (normalized breakdown point) for general non-symmetric distributions can be as low as $O(1/d)$. See, for example, the discussions in [DM92, CT02, ZJS20] on asymmetric distributions. The class of distributions we consider includes non-symmetric distributions. Therefore, the Tukey median for our class of distributions inherently fails.
> > > > >
> > > > > ---
> > > > > +  [DM92] D. L. Donoho, M. Gasko. 1992. Breakdown Properties of Location Estimates Based on
> > > > > Halfspace Depth and Projected Outlyingness
> > > > > + [CT02] Z. Chen, D. Tyler. 2002. The Influence Function and Maximum Bias of Tukey's Median
> > > > > + [ZJS20] B. Zhu, J. Jiao, J. Steinhardt. 2020. When does the Tukey Median work?

---

### Official Review · Reviewer_DNt7 · 2022-07-11

**Rating:** 5
**Confidence:** 1
**Soundness:** 3 good
**Presentation:** 3 good
**Contribution:** 3 good

**Summary:**

This paper considers robust mean estimation problem with sparsity constraint for heavy-tailed distribution. The proposed estimator is claimed to achieve the first sample complexity which has a logarithmic dependence on the dimension and tightly matches with other parameters.

**Questions:**

N/A

**Limitations:**

Yes.

**Strengths And Weaknesses:**

I appreciate authors' effort in the paper. But to be honest, this paper is out of my expertise, and I found it hard for me to write a review which makes some sense. The system also keeps failing to process my request for a change in reviewers. To keep a high quality of review, I will just make an educational guess, authors and other reviewers may just skip my review.

Thank you for the effort.

Reviewer

---

### Official Review · Reviewer_5B6A · 2022-07-18

**Rating:** 4
**Confidence:** 4
**Soundness:** 3 good
**Presentation:** 2 fair
**Contribution:** 2 fair

**Summary:**

This paper studies outlier-robust sparse mean estimation for heavy-tailed distributions. Prior work has considered robust sparse mean estimation for the light-tailed distribution (such as sub-Gaussian). The sample complexity for the proposed algorithm scales with k^2 log d, which is near optimal, and the assumption for the underlying heavy tailed distribution is mild.

However, the technical contribution of this paper is incremental: the usage of coordinate MOM estimators for heavy tailed distribution is standard, and the filtering analysis basically follows from [BDLS17], which involves a time-consuming sparse PCA relaxation. Hence, given the theoretical guarantees of prior algorithmic result, the remaining challenge is to show that a large subset of the uncorrupted heavy tailed samples satisfies the “stability” condition.

**Questions:**

The presentation of Algorithm 1 is confusing. It would be great if authors can present the algorithmic box in a self-consistent fashion.

**Limitations:**

Please refer to the Strengths And Weaknesses session.

**Strengths And Weaknesses:**

Strength:

-- The introduction of the related work is comprehensive, and the theoretical presentation is clear.
-- The theoretical guarantees matches the best-known results.

Weakness:

-- The theoretical innovation is quite limited.
-- The actual running time for the algorithm is very time consuming compared to other robust sparse mean estimators.
-- Provided vast literatures in robust sparse mean estimation ([BDLS17, DKK+19]), this paper does not have experimental comparisons to show potential superior performance than these baselines.

---

> ### Author Response · Authors · 2022-08-01
> **Authors' response to Reviewer 5B6A**
>
> We thank the reviewer for their time and effort in reviewing our paper, and for the feedback. In terms of the presentation of Algorithm 1, we would appreciate additional concrete points of confusion, so that we can improve the presentation of the algorithm.
>
> Our responses to the questions raised in the review:
>
> + Our technical contribution: As we mentioned in the overall response, we agree that the algorithmic component in the paper is not the main contribution – we do not claim any credit for using coordinatewise median-of-means or the sparse filtering algorithm of [BDLS17]. The innovation in this paper lies in the highly challenging analysis showing the existence of a large stable subset of samples with sub-Gaussian rate (Section 5 in the paper), which led to our novel non-spectral decomposition of sparse positive semi-definite matrices. $\qquad\qquad\qquad\qquad\qquad\qquad\qquad\qquad\qquad\qquad\qquad\qquad\qquad\qquad\qquad\qquad\qquad\qquad\qquad\qquad\qquad\qquad\qquad\qquad$
> + Regarding “*The theoretical guarantees matches the best-known results*”  and “*Provided vast literatures in robust sparse mean estimation ([BDLS17, DKK+19]), this paper does not have experimental comparisons to show potential superior performance than these baselines*”
>
>   We want to emphasize again that there is no prior work at all in the setting we consider in the paper. All previous poly-time robust sparse mean estimators [BDLS17, DKK+19] assume light-tails, and as stated on line 224 and shown by Example 1, these algorithms provably fail. Our algorithm is the first theoretical result in the heavy-tailed setting, and therefore there is no prior work to compare with.

---

### Official Review · Reviewer_L8yB · 2022-08-11

**Rating:** 6
**Confidence:** 4
**Soundness:** 3 good
**Presentation:** 3 good
**Contribution:** 2 fair

**Summary:**

The paper considers the problem of robust sparse mean estimation. In this problem, we observe samples from a heavy-tailed distribution whose mean is a sparse vector. This distribution is assumed to have bounded variance and is assumed to satisfy a weaker form of bounded 4th moment assumption. In addition, it is assumed that a constant fraction of the samples are arbitrarily corrupted by a malicious adversary. For this problem, the authors propose a mean estimator which achieves convergence rates of the order $O(\sqrt{\epsilon} + \sqrt{k^2\log{d}/n} + \sqrt{\log(1/\tau)/n})$, where epsilon is the fraction of corruptions, k is the sparsity, d is the dimension, tau is the failure probability. The algorithm is based on the popular filtering algorithm and involves an additional truncation step for ensuring the samples satisfy the stability condition.

**Questions:**

See above

**Limitations:**

See above

**Strengths And Weaknesses:**

First of all, I have been asked to review the paper at the last minute. So I didn't have time to read the proofs in the appendix. But, I have gone through the main paper.

The paper addresses two corruption models simultaneously: malicious adversary model and heavy-tailed model. In addition, the paper tries to do sparse mean estimation. The heavy-tailed part in the paper looks interesting and could be useful to the community (especially given that there is no prior work which can get computationally efficient estimators for this problem). However, it is not immediately clear if the results for malicious adversaries are optimal, and is debatable (see below for more details). The core algorithmic ideas are not novel (except for the truncation step). The main novelty is probably in the analysis of the truncation step. Nonetheless, I believe the pros of the paper marginally outweigh the cons.

$$\textbf{Theoretical Results}.$$
1. $\epsilon\  \textbf{dependence}.$  It is not clear if the derived error bounds have optimal dependence on $\epsilon$. For bounded $4^{th}$ moment distributions, the optimal rates have $\epsilon^{3/4}$ dependence [1,3]. I understand that the paper considers a weaker condition than bounded $4^{th}$ moment. But the lower bounds presented in the paper don't clearly show what the optimal dependence should be. The current lower bounds only hold for $k>\epsilon^{-1/2}.$  For smaller $k$, will the lower bounds still be $\epsilon^{1/2}$? At least for the hard instance created in the proof, it looks like the lower bounds will be $\epsilon^{3/4}$ when $k=1$. Given this, it is hard to judge if the rates are optimal in $\epsilon$ for all $k$.
2. $4^{th}$ $\textbf{moment condition}$. The $4^{th}$ moment condition in the paper looks weird. It is not a rotational invariant condition. Let's say I have a distribution which satisfies this condition. Now, I create another distribution by rotating this distribution such that the new distribution doesn't satisfy the moment condition. The algorithm works for the first distribution but not the second distribution. This naturally raises the question if this assumption can be further relaxed or totally removed. It'd be great if the authors try to relax this condition.

$$\textbf{Other comments}.$$
1. Some relevant work hasn't been cited [1,2]. The current work is certainly not the first to study sparse mean estimation in heavy-tailed settings with outliers. [1] considers the same problem as the one considered in this work and provides an optimal estimator, albeit they provide a computationally inefficient estimator. These works should cited and discussed in the paper.
2. At a number of places it is claimed that for robust mean estimation of Gaussian distribution, the error rates will scale as $O(\sqrt{\epsilon})$ (for example, Fact 1.2). This is not correct and needs to be fixed. The optimal error rates will scale as $O(\epsilon)$ [3].
3. Why is $||\cdot||\_{2,k} $ norm used instead of $||\cdot||\_{2}$ norm, if they are both equivalent upto constant factors?

[1] A Robust Univariate Mean Estimator is All You Need. https://proceedings.mlr.press/v108/prasad20a.html

[2] A Unified Approach to Robust Mean Estimation. https://arxiv.org/pdf/1907.00927.pdf

[3] Recent Advances in Algorithmic High-Dimensional Robust Statistics. https://arxiv.org/pdf/1911.05911.pdf


-------------------------------------------------------------------------------------------------------------------------------------------
Update after discussion with other reviewers: I believe the heavy tailed results in the paper are useful, especially given that this is the first computationally efficient estimator in this setting. So I'm increasing my score.

---

### Author Response · Authors · 2022-08-01
**Overall Response**

We thank the reviewers for the detailed feedback. We will address your questions and remarks by responding directly to each review, and here at a top level comment we wish to highlight and emphasize the following few points about our contributions.

1. The robust+sparse+heavy-tailed mean estimation problem **previously had no polynomial-time estimators** that achieve any dimension-independent error (even with constant probability). Our result is the **first** such result in the literature. Other (algorithmic) works in robust sparse mean estimation require far stronger assumptions such as (sub-)Gaussianity. Moreover, all prior robust sparse mean polytime estimators *provably fail* without the crucial (though simple) truncation step we introduce (see line 224 and Example 1). As such, there is no prior work for us to directly compare with, either theoretically or experimentally.
2. From the lack of prior work, our results achieve not only polynomial dependence in both time and (in fact, optimal under a standard conjecture) sample complexity, but we also achieve *optimal dependence on the failure probability*, enabling the best possible rates in the high probability regime.
3. The main technical innovations of this paper indeed lie in our analysis, which is a **highly non-trivial** instantiation of the stability-based framework. Our novel analysis importantly yields new tools for understanding and characterizing distributions with sparse mean vectors, which may be useful for other robust sparse estimation tasks. We stress again that we do not claim any innovation on the algorithmic front.
4. To be more explicit, our rounding scheme significantly differs from [DL22]. First, our decomposition is (necessarily) non-spectral and decomposes a PSD matrix with bounded $\ell_1$ norm into sparse matrices with bounded Frobenius norm
(the set
$\mathcal{A}\_k$
 in Equation (3)).
Second, these component sparse matrices are further constructed to tailor to the underlying inlier data distribution (the set $\mathcal{A}_{k,P}$ defined in Equation (3)). The construction is intricate, and we believe that it is an interesting and novel structural result that could be useful for further work on the problem.

---

### Meta-Review · Area_Chair_hR9z · 2022-08-29

**Recommendation:** Accept
**Confidence:** Certain

**Metareview:**

The paper studies the problem of sparse mean estimation in presence of heavy tailed noise. The result of the paper are novel and  interesting. Improvement over work of Prasad et al is significant as the paper is able to provide a computationally efficient method for the problem. Claims about Huber contamination model should be toned down, as it is not clear if the result is optimal.

The paper generated a fair bit of discussion, and as mentioned above there are concerns about some of the claims of the paper, and they should be toned down or should explicitly highlight issues like dependence on epsilon.
Nonetheless, reviewers agree that the paper represents significant improvement in SOTA owing to computationally efficient method.

**Award:**

No

---

### Decision · Program_Chairs · 2022-09-14

Accept